

**Seasonal characteristics of emission, distribution, and radiative effect of marine organic aerosols**
**over the western Pacific Ocean: an investigation with a coupled regional climate-aerosol model**
Jiawei Li[1], Zhiwei Han[1,2*], Pingqing Fu[3], Xiaohong Yao[4]
[1]CAS Key Laboratory of Regional Climate-Environment for Temperate East Asia (RCE-TEA),
Institute of Atmospheric Physics, Chinese Academy of Sciences, Beijing 100029, China
[2]University of Chinese Academy of Sciences, Beijing 100049, China
[3]School of Earth System Science, Tianjin University, Tianjin 300072, China
[4]Laboratory of Marine Environmental Science and Ecology, Ministry of Education, Ocean University
of China, Qingdao 266100, China
Correspondence to: Zhiwei Han (hzw@mail.iap.ac.cn)
**Abstract**: Organic aerosols from marine sources over the western Pacific Ocean of East Asia were
investigated by using an online-coupled regional chemistry-climate model RIEMS-Chem for the
entire year 2014. Model evaluation against a wide variety of observations from research cruises and
in-situ measurements demonstrated a good skill of the model in simulating temporal variation and
spatial distribution of particulate matter with aerodynamic diameter less than 2.5 μm and 10 μm
($PM_{2.5}$ and $PM_{10}$), black carbon (BC), organic carbon (OC), sodium, and aerosol optical depth (AOD)
in the marine atmosphere. The inclusion of marine organic aerosols apparently improved model
performance on OC aerosol concentration. The regional and annual mean near surface marine organic
aerosol (MOA) concentration was estimated to be 0.27 μg m$^{-3}$, with the maximum in spring and the
minimum in winter and contributed 26% of the total organic aerosol concentration on average over the
western Pacific. Marine primary organic aerosol (MPOA) accounted for the majority of marine
organic aerosol (MOA) mass and exhibited the maximum in autumn and the minimum in summer,
whereas marine secondary organic aerosol (MSOA) was approximately 1~2 orders of magnitude
lower than MPOA, having a distinct summer maximum and a winter minimum. MOA induced a direct
radiative effect ($DRE_{MOA}$) of -0.27 W m$^{-2}$, and an indirect radiative effect ($IRE_{MOA}$) of -0.66 W m$^{-2}$ at



TOA ($IRE_{MOA}$) in terms of annual and oceanic average over the western Pacific, with the highest seasonal mean $IRE_{MOA}$ up to -0.94 W m$^{-2}$ in spring. $IRE_{MOA}$ was stronger than but in a similar magnitude to the IRE due to sea salt aerosol on average, and it was approximately 9% of the IRE due anthropogenic aerosols in terms of annual mean over the western Pacific, and this ratio increased to 19% in the northern parts of the western Pacific in autumn. This study reveals an important role of MOA in perturbing cloud properties and shortwave radiation fluxes in the western Pacific of East Asia.

## 1 Introduction

Atmospheric aerosol is one of the most important and uncertain factors in climate change issues (IPCC, 2013). Aerosols can alter radiation balance by scattering/absorbing solar/infrared radiation, and affect cloud microphysics and lifetime by activating as cloud condensation nuclei (CCN), exerting significant effects on climate system directly and indirectly. Aerosols are originated from anthropogenic and natural sources and of high spatial and temporal variability and short atmospheric lifetime relative to greenhouse gases. Consequently, aerosol radiative and climatic effects often have strong regional characteristics.

The western Pacific Ocean is frequently influenced by continental outflow of both anthropogenic and natural aerosols. Due to continuous growth of economy and energy consumption in the past decades, the aerosol level in China has been enhanced (Smith et al., 2011; Li M. et al., 2017) and may have potentially significant effects on radiation and cloud over not only the East Asian continent but also the wide downwind oceanic areas. Besides, East Asia is one of the major dust source regions on earth (Shao and Dong, 2006). Dust storms often occur in spring and dust particles can be transported eastward from the deserts and Gobi areas of north China and southern Mongolia to the western Pacific Ocean (Gong et al., 2003), providing nutrients (e.g., iron) for phytoplankton or even triggering the outbreak of algae bloom in oceans (Calil et al., 2011; Tan et al., 2017). In addition to anthropogenic and dust aerosols, marine aerosols also significantly affect aerosol chemical composition, radiation transfer, and cloud properties in marine atmosphere. The behaviors and climatic impacts of sea salt and non-sea-salt sulfate oxidized from dimethylsulphide (DMS) have been extensively investigated (Graf et al., 1997; Liao et al., 2004; Rap et al., 2013). In recent years, particular attentions have been paid on the sources and impacts of marine organic aerosols (O'Dowd et al., 2004; Meskhidze and Nenes, 2006; Luo and Yu, 2010; Vignati et al., 2010; Gantt et al., 2011; Burrow et al., 2014; Quinn et





al., 2017; Bertram et al., 2018; Huang et al., 2018), however, such studies were still very limited,
especially for the western Pacific.
O'Dowd et al. (2004) found that organic matter dominated the chemical composition of marine
aerosol during plankton bloom periods from spring to autumn over the North Atlantic Ocean,
contributing 63% to sub-micron aerosol mass. Meshkidze and Nenes (2006) revealed a significant
impact of phytoplankton bloom on cloud droplet number concentration and radiation balance in the
Southern Ocean and proposed a major contribution of secondary organic aerosol (SOA) from
phytoplankton produced isoprene. Some studies indicated that primary marine sources may dominate
marine organic matter, whereas SOA oxidized from marine isoprene could only comprise a small
fraction of the observed organic aerosol mass over marine environment (Facchini et al. 2008; Arnold
et al., 2009; Myriokefalitakis et al., 2010). The estimated global emission amounts of primary marine
organic matter varied largely among models. Using the global aerosol-climate model ECHAM5-HAM,
Roelofs (2008) estimated a global production of marine organic aerosols to be 75 TgC yr$^{-1}$. Spracklen
et al. (2008) estimated the marine organic carbon emission to be approximately 8 TgC yr$^{-1}$ based on
measured organic carbon mass and satellite retrieved chlorophyll-a (Chl-a) concentration. Vignati et al.
(2010) derived a global emission of marine primary organic matter in the sub-micron size by sea spray
process to be 5.8 TgC yr$^{-1}$ by using an off-line global Chemistry-Transport Model TM5 with a
parameterization relating organic emission fraction to sea surface Chl-a concentration. Gantt et al.
(2011) found that the combination of 10 m wind speed and sea surface Chl-a concentration were the
most consistent predictors of organic mass fraction of sea spray aerosol based on observations from
the Mace Head atmospheric research station on the Atlantic coast of Ireland and a site at the Point
Reyes National Seashore on the Pacific coast of California. They developed a new MPOA emission
function and estimated the global annual MPOA emission associated with sea spray to be from 15.9
TgC yr$^{-1}$ to 18.7 TgC yr$^{-1}$ (2.8~5.6 TgC yr$^{-1}$ in the sub-micron size). However, Quinn et al. (2014)
found that the organic carbon content of sea spray aerosol is weakly correlated with satellite retrieved
chlorophyll-a concentration based on cruise measurements in the North Atlantic Ocean and the coastal
waters of California. Bates et al (2020) reported that plankton bloom has little effect on the emission
flux, organic fraction or cloud condensation nuclei of sea spray aerosol based on cruise experiment
over the North Atlantic. Burrows et al. (2014) developed a novel physically based framework for
parameterizing the organic fractionation of sea spray aerosol by consideration of ocean



biogeochemistry processes, and their predicted relationships between Chl-a and organic fraction are
similar to existing empirical parameterizations associated with ocean Chl-a concentrations at high
Chl-a levels, but the empirical relationships may not be adequate to predict OM fraction of sea spray
aerosol outside of strong seasonal blooms. Considering the strong bloom seasonality in the western
Pacific region and the availability of global satellite data for Chl-a concentration, and the lack of
cruise measurements on the relationship between sea spray organic aerosol fluxes and Chl-a in this
region, we adopted the scheme of Gantt et al (2011) for parameterizing marine primary organic
aerosol emission in this study.
Regarding the influence on climatic factors, such as cloud condensation nuclei (CCN),
Ovadnevaite et al. (2011) revealed that MPOA was a dichotomy of low hygroscopicity and high CCN
activity through analysis of ambient measurements of aerosol chemical compositions and size
distributions at the Mace Head atmospheric research station, and highlighted the importance of MPOA
in CCN activation over marine atmosphere. A later study of Westervelt et al. (2012) indicated that
marine organic aerosols was able to increase CCN by up to 50% in the Southern Ocean and by 3.7%
globally during the austral summer based on the model simulation of GISS GCM II'. Based on the
measurements from seven research cruises over the Pacific, Southern, Arctic, and Atlantic oceans
between 1993 and 2015, Quinn et al. (2017) indicated that sea spray aerosol generally makes a
contribution of less than 30% to CCN population at supersaturation of 0.1 to 1.0% on a global basis.
Burrows et al. (2022) pointed out that sea spray organic aerosol strengthened shortwave radiative
cooling by clouds by $-0.36$ $Wm^{-2}$ in the global annual mean, with the zonal mean contribution
exceeding -3.5 W m$^{-2}$ in the Southern Ocean in summertime.
The above studies reveal the important role of marine organic aerosols in chemical composition,
radiation budget, and cloud microphysics with focus on the global scale. However, there is very
limited modeling research on this important and challenging issue for the western Pacific Ocean of
East Asia. To our knowledge, only two of our previous studies explored the effects of MPOA on
chemical composition, radiation, cloud and precipitation over the western Pacific in springtime with
an online-coupled regional chemistry/aerosol-climate model RIEMS-Chem (Han et al., 2019; Li et al.,
2019), whereas the seasonality and annual aspect of MPOA and MSOA produced by marine isoprene
and terpene are still unknown. In this study, we conducted a one-year simulation with the developed
RIEMS-Chem to further explore the characteristics and radiative impacts of marine organic aerosols



over the western Pacific. The model simulated aerosol compositions were validated against a wide series of observations from ground and cruise measurements, and the simulated MSOA was evaluated by comparison with cruise measured secondary organic tracer in marine air masses. To our knowledge, for the first time, the seasonality of emissions, concentrations, direct and indirect radiative effects of marine organic aerosols was characterized and the annual means were estimated specifically for the western Pacific and for the key oceanic regions of concern over East Asia. This study would provide new insights into properties and impacts of marine organic aerosols over the western Pacific and would be a necessary supplement to the global perspective of marine organic aerosols.

## 2 Model and data

### 2.1 Model description and key processes

An online-coupled regional atmospheric chemistry/aerosol-climate model RIEMS-Chem was used to investigate marine organic aerosols in this study. RIEMS-Chem composes of the host regional climate model RIEMS (Fu et al., 2005; Xiong et al., 2009; Wang S.Y. et al., 2015) and a comprehensive atmospheric chemistry/aerosol module. RIEMS was developed based on the dynamic structure of the fifth-generation Pennsylvania State University NCAR Mesoscale Model (MM5; Grell et al., 1995) with a series of parameterizations to represent major physical processes, such as a modified Biosphere-Atmosphere Transfer Scheme (BATS; Dickinson et al., 1993) for land-surface process, the Medium-Range Forecasts scheme (MRF; Hong and Pan, 1996) for planetary boundary layer process, the Grell cumulus convective parameterization scheme (Grell, 1993) for convective process, the Reisner explicit moisture scheme (Reisner et al., 1998) and a modified radiation package of the NCAR Community Climate Model (CCM3; Kiehl et al., 1996) for radiation transfer processes with aerosol effect. RIEMS has participated in the Regional Climate Model Intercomparison Project (RMIP) for Asia and it was one of the best models in predicting surface air temperature and precipitation over East Asia (Fu et al., 2005).

Atmospheric chemistry/aerosol modules have been incorporated into RIEMS in recent years, establishing the online-coupled model RIEMS-Chem, which can account for the interactions among chemistry, radiation, cloud, and meteorology (Han, 2010; Han et al., 2012). RIEMS-Chem has been successfully applied in previous modeling studies on anthropogenic aerosols, mineral dust and marine aerosols regarding spatial-temporal distributions, physical and chemical evolutions, and radiative and



climatic effects over East Asia (Han et al., 2012; 2013; 2019; Li et al., 2014; 2016a; 2016b; 2019;
2020). It is now participating in the international model comparison project MICS-Asia III (Model
Inter Comparison Study for Asia phase III) and shows a good ability in predicting aerosol
concentrations and AOD over East Asia (Gao et al., 2018).
RIEMS-Chem includes atmospheric chemistry and aerosol processes, such as gas and aqueous
phase chemistries which are represented by the CB-IV mechanism (Gery et al. 1989) and RADM
scheme (Chang et al., 1987), respectively; Sulfate is mainly produced from the oxidation of $SO_2$ by
OH radical in gas phase and the oxidation of dissolved $SO_2$ by $H_2O_2$, $O_3$, and metal catalysis in
aqueous phase (Chang et al., 1987). Nitrate and ammonium are produced through thermodynamic
processes represented by the ISORROPIA II model (Fountoukis and Nenes, 2007). BC, POA, and
anthropogenic primary PMs are considered chemically inert. SOA formation from anthropogenic and
biogenic VOC precursors is treated by a bulk yield scheme from Lack et al. (2004), with SOA yield of
424 $\mu g\ m^{-3}\ ppm^{-1}$ for toluene, 342 $\mu g\ m^{-3}\ ppm^{-1}$ for xylene, and 762 $\mu g\ m^{-3}\ ppm^{-1}$ for monoterpene.
For irreversible conversion of marine VOCs to SOA, a 28.6% mass yield is assumed for isoprene
(Surratt et al., 2010, Meskhidze et al., 2011) and 30% for monoterpene (Lee et al., 2006).
Heterogeneous reactions between gaseous precursors and aerosols are also taken into account (Li and
Han, 2010; Li J. W. et al., 2018). Dry deposition velocity is represented by a size-dependent
parameterization over different underlying surfaces (Han et al., 2004). Dry deposition velocity of
particle is expressed as the inverse of the sum of resistant plus a gravitational settling term. Over sea
or ocean surfaces, the quasi-laminar boundary layer (QBL) may be disrupted by bursting bubbles,
resulting in an increase in downward movement of particles, which is parameterized by the approach
of Van den Berg et al. (2000), in which quasi-laminar resistance $r_b$ is determined by Brownian
diffusion and impaction when QBL is intact, and by turbulence and washout velocity of particles by
spray drops when QBL is broken down. Below-cloud scavenging (BCS) of particles between cloud
base and ground surface represents capture processes of particle by falling hydrometeor through
Brownian and turbulent shear diffusion, interception and inertial impaction, and is parameterized by a
scavenging rate, which is a function of precipitation rate and collision efficiency of particle by
hydrometeor (Slinn, 1984).
A physically based scheme (namely A-G scheme) developed based on classical Köhler theory by
Abdul-Razzak and Ghan (1998, 2004) is incorporated into RIEMS-Chem to represent aerosol





activation into cloud droplet processes. This scheme calculates cloud droplet number concentration
($N_c$) with not only aerosol number concentration, but also aerosol size distribution and composition,
updraft velocity and ambient supersaturation. Aerosols are activated if their critical supersaturation is
less than the maximum ambient supersaturation. The critical supersaturation for activating particles is
determined by curvature effect and solute effect. The maximum ambient supersaturation is calculated
by solving supersaturation balance equation (Abdul-Razzak and Ghan, 1998). The updraft velocity is
represented by the sum of grid mean updraft velocity and sub-grid updraft velocity, which is
diagnosed from vertical eddy diffusivity according to Ghan et al. (1997). The A-G scheme in
RIEMS-Chem has been applied over the western Pacific Ocean in spring 2014 and its prediction for
hourly CCN concentration at different supersaturations has been validated by cruise measurements
from the marginal seas of China to remote oceans southeast of Japan, which demonstrates a good
ability, with the correlation coefficient of 0.87 and normalized mean bias within 20%. More details on
the treatment and evaluation of marine aerosol activation refer to Han et al. (2019). Once $N_c$ is
calculated by the A-G scheme, the cloud droplet effective radius $r_e$ is calculated with the method of
Martin et al. (1994). The activated aerosols (into cloud droplet) are removed from the air. The auto
conversion rate from cloud water to rainwater is parameterized by the scheme of Beheng (1994),
which depends on $N_c$ diagnosed and cloud liquid water content. The effect of aerosols on ice nuclei
and convective cloud is not treated in this model due to limited knowledge at present.

2.2 Aerosol physical and chemical properties
Ten aerosol types are simulated in RIEMS-Chem, which are sulfate ($SO_4^{2-}$), nitrate ($NO_3^-$),
ammonium ($NH_4^+$), black carbon (BC), primary organic aerosol (POA), secondary organic aerosol
(SOA), anthropogenic primary PMs ($PM_{2.5}$ and $PM_{10}$), dust, and sea salt.
Based on the observational analysis of aerosol mixing state in eastern China (Ma et al., 2017; Wu
et al., 2017), an internal mixing assumption is adopted for anthropogenic aerosols and they are
externally mixed with natural aerosols. The geometric mean radius and standard deviation of the
anthropogenic internal mixture are estimated to be 0.11 µm and 1.65, respectively, based on field
measurements (Ma et al., 2017). Mineral dust and sea salt are represented by 5 size bins (0.1~1.0,
1.0~2.0, 2.0~4.0, 4.0~8.0 and 8.0~20.0 µm). Feng et al. (2017) indicated that the measured total
organic carbon (TOC) in the western Pacific Ocean during the same period as this study was enriched



in <0.3 μm (volume median diameter), which was mainly contributed by MPOA and the super-micron
TOC was generally below the detection limit. Accordingly, the geometric mean diameter of marine
organic aerosol number concentration was set to be 0.1 μm, with a standard deviation of 1.6. The
number concentration is calculated by mass concentration as the formula in Curci et al. (2015). MPOA
can be mixed with sea salt both externally or internally. It is more likely to be externally mixed with
sea salt for finer aerosols (<200 nm in diameter) (Gantt and Meskhidze, 2013) and the effect of
externally mixed MPOA was found to be much more important than that of internally mixed MPOA
(Gantt et al., 2012b). So an external mixture of MPOA and sea salt is assumed in this study, this
means additional marine organic aerosols are produced to affect cloud properties and represents an
upper limit of indirect effect. There is little information on physical and chemical properties of marine
organic aerosols, some key parameters for calculation of aerosol activation, i.e. the number of ions the
salt dissociates into water, the osmotic coefficient, the mass fraction of soluble material, the density,
and molecular weight are set to be 3.0, 1, 0.1, 1.5 g m$^{-3}$, and 90, respectively, according to a few
previous studies (Abdul-Razzak and Ghan, 2004; Roelofs, 2008). The soluble mass fraction of MSOA
is assumed to be 0.2, slightly higher than that of MPOA (Liu and Wang, 2010; Westervelt et al., 2012).
An OM/OC ratio of 1.4 was applied to convert organic matter (OM) to OC (Gantt and Meskhidze,

2013).

The hygroscopic growth of aerosol is parameterized by a κ parameterization (Petters and
Kreidenweis, 2007). The hygroscopicity parameters (κ) for inorganic aerosol components, BC, POA,
SOA, dust, and sea salt are set to be 0.65, 0, 0.1, 0.2, 0.01 and 0.98, respectively (Riemer et al., 2010;
Liu et al., 2010; Westervelt et al., 2012). The hygroscopicity for MPOA and MSOA are assumed to be
the same as those for anthropogenic POA and SOA. Because there was limited information on the
optical properties of marine organic aerosols, the refractive index of anthropogenic POA and SOA was
used instead. The aerosol refractive index and hygroscopicity (κ) of the internally mixed aerosol are
calculated by volume-weighting of the parameters for each aerosol component. Aerosol optical
parameters including extinction coefficient, single scattering albedo, and asymmetry factor are
calculated by a Mie-theory based method developed by Ghan and Zaveri (2007), in which the aerosol
optical parameters are pre-calculated by the Mie theory and then fitted by Chebyshev polynomials
with a table of polynomial coefficients for looking up for aerosols with certain size and refractive
index. More detailed description refers to Li et al. (2020). This approach is much faster than



traditional Mie code with a similar level of accuracy and has been successfully used in estimating
aerosol optical properties over East Asia (Han et al. 2011).

2.3 Anthropogenic and natural emissions

Monthly mean anthropogenic emissions of sulfur dioxide ($SO_2$), nitrogen ($NO_x$), ammonia ($NH_3$),

non-methane volatile organic compounds (NMVOC), carbon monoxide (CO), BC, POA, and other
anthropogenic primary $PM_{2.5}$ and $PM_{10}$ in China for the year 2014 are obtained from the MEIC
inventory (Multi-resolution Emission Inventory for China) which was developed by Tsinghua
University (http://meicmodel.org, last access: 2020/01/20). Anthropogenic emissions outside China
are taken from the MIX inventory which was developed to support the Model Inter-Comparison Study
for Asia phase III (MICS-Asia III) and the Hemispheric Transport of Air Pollution (HTAP) projects
(Li M. et al., 2017). Both inventories of MEIC and MIX have the same resolution of 0.5 degree. Open
biomass burning emissions of aerosols and gas precursors for the year 2014 with a spatial resolution
of 0.5 degree are derived from the Global Fire Emissions Database, Version 4.0 (GFED4) on a daily
basis (Giglio et al., 2013). The biogenic VOC emission is derived from the CAMS-BIO Global
biogenic emissions dataset (CAMS-GLOB-BIO v3.1) (Granier et al., 2019; Sindelarova et al., 2014)
distributed by ECCAD-GEIA (https://permalink.aeris-data.fr/CAMS-GLOB-BIO, last access:
2020/02/10) and the monthly mean biogenic emission for the year 2014 with a horizontal resolution of
0.25° is used. All the above emission data are bilinearly interpolated to the lambert projection of
RIEMS-Chem. The deflation of mineral dust is represented by the scheme of Han et al. (2004), which
is calculated online with RIEMS predicted meteorology.

The generation of sea salt aerosol through bubbles is represented by Gong (2003) which is

developed for sea salt radius from 0.07μm to 20μm based on the scheme of Monahan et al. (1986) and
it is modified by considering the influences of sea surface temperature (SST) (Jaeglé et al., 2011) and
relative humidity (RH) (Zhang et al., 2005).

The size-resolved marine primary organic aerosol (MPOA) emission is parameterized based on

the method of Gantt et al. (2011; 2012a), in which the emission rate of MPOA is the product of sea
spray emission rate and organic matter fraction of sea spray aerosol, which is expressed as a function
of wind speed, surface seawater Chl-a concentration, and aerosol size. The Chl-a data used in this
study is the Level-3 daily mean Chl-a concentration (mg m$^{-3}$) product with 9 km resolution retrieved



from the VIIRS (Visible infrared Imaging Radiometer) sensor onboard the Suomi National
Polar-orbiting Partnership (SNPP) satellite platform (OBPG, 2018) (http://oceandata.sci.gsfc.nasa.gov,
last access: 2020/12/8). A brief description on this scheme with formulas is presented in the
supplement.
Marine isoprene emission released by phytoplankton activities is parameterized using the scheme
of Gantt et al. (2009) which considers light sensitivity of phytoplankton isoprene production and
dynamic euphotic depth (see more details in the supplement). Marine emission of monoterpene is
scaled by 0.2 to those of isoprene following the suggestion from Myriokefalitakis et al. (2010). The
marine abiotic source of isoprene (due to photochemical production in the sea surface microlayer)
may be important according to recent studies (Brüggemann et al., 2018; Conte et al., 2020). This
mechanism is not considered in this study because the production mechanism for marine abiotic
isoprene is poorly understood at present.

2.4 Model setup and experiment design
This study focused on the western Pacific Ocean of East Asia. The model domain covered most
areas of eastern China, the Korean Peninsula, Japan, parts of Southeast Asia, and a wide area of the
western Pacific Ocean (Figure 1). A lambert conformal projection with 60 km horizontal resolution
was applied in the model. 16 vertical layers stretched unevenly from the surface to tropopause in a
terrain-following sigma coordinate with the first 8 layers within planetary boundary layer. The
simulation period was from 1 December 2013 to 31 December 2014 with the first month as model
spin-up and the whole year of 2014 was used for analysis. Final reanalysis data with 1°×1° resolution
and 6-hour interval from the National Centers for Environmental Prediction (NOAA/NCEP, 2000)
was used to provide initial and boundary conditions for meteorology. Chemical results derived from
the MOZART-4 (Model for Ozone and Related chemical Tracers, version 4; Emmons et al., 2010)
simulation with 6-hour interval were used to provide lateral conditions for trace gases and aerosols.
The full simulation (FULL) is designed by considering all anthropogenic and natural emissions,
including marine emissions of primary organic aerosol and sea salt, and a series of model simulations
are also conducted to estimate the direct and indirect radiative effect of MOA and their sensitivity to
MOA properties, which are described in the following sections.



2.5 Observations
In-situ measurements of PM$_{10}$, PM$_{2.5}$, and gas precursors (O$_3$, SO$_2$, and NO$_x$/NO$_2$) at coastal and
island sites in Japan and Republic of Korea were obtained from EANET (Acid Deposition Monitoring
Network in East Asia, http://www.eanet.asia, last access: 2020/01/23) (Figure 1). Hourly
concentrations of PM$_{10}$, SO$_2$, NO$_x$ in Japan, NO$_2$ in Korea, and O$_3$ were automatically monitored at
six Japanese sites (Rishiri, Tappi, Sado-seki, Oki, Hedo, and Ogasawara) and three Korean sites (Jeju,
Kanghwa, and Imsil), whereas hourly PM$_{2.5}$ concentrations were only available at three Japanese sites
(Rishiri, Sado-seki, and Oki). Sodium (Na$^+$) concentrations sampled on a bi-weekly basis at the 6
coast/island EANET sites in Japan were also collected. Besides, hourly PM$_{10}$ and PM$_{2.5}$
concentrations monitored in the three coastal cities of China (Qingdao, Shanghai, and Fuzhou) were
also    obtained    from    the    CNEMC    (China    National    Environmental    Monitoring    Center,
http://www.cnemc.cn/, last access: 2020/01/23) and used for model validation (Figure 1).
Carbonaceous aerosol (OC and BC) concentrations measured from two research cruise campaigns
covering the western Pacific during the spring and summer of 2014 (Figure 1) were collected and used
for model validation. The spring cruise campaign was carried out from 17 March to 22 April 2014
onboard the research vessel R/V Dongfanghong II, which started from Qingdao, sailed to the western
Pacific Ocean, and then returned (Figure 1) (Luo et al., 2016; Feng et al., 2017). OC and BC samples
were    collected    by    an    11-stage    MOUDI    (Models110-IITM)    (0.054~18    μm)    equipped    with
pre-combusted quartz filters onboard the vessel. Mass concentrations of total OC (primary and
secondary) and BC were determined by the thermal/optical carbon analyzer (Sunset Laboratory Inc.,
Forest Grove, OR). Totally 19 daily BC and OC samples were collected during the cruise. Detailed
information about this campaign and the sampling and analysis techniques were documented in Feng
et al (2017). The early summer campaign was carried out from 18 May to 12 June 2014 (Kang et al.,
2018). Total suspended particles (TSP) were collected on pre-combusted quartz filters using a
high-volume air sampler (Kimoto, Japan) onboard the KEXUE-1 Research Vessel during a National
Natural Science Foundation of China (NSFC) sharing cruise (Figure 1). This campaign covered low-
to mid-latitudes of the western Pacific Ocean (over the Yellow Sea and the East China Sea). Totally 51
half-day (daytime/nighttime) OC samples were obtained during this campaign. Detailed information
about this campaign and samples were described in Kang et al. (2018).
Besides the in-situ observations and cruise campaigns mentioned above, long-term observations



of OC and BC from previous publications were collected to help model comparison and analysis.
Carbonaceous aerosol samples (OC and BC) in TSP were continuously collected on a weekly basis
from 2001 to 2012 at Chichijima Island (the same place as Ogasawara in Figure 1), a remote island
located in the western North Pacific. The monthly mean OC and BC concentrations of the 12-year
average were reported by Boreddy et al. (2018) and used to verify the model performance over remote
oceans. Measurements of seasonal mean OC and BC concentrations in TSP at Huaniao Island (a
pristine island about 100 km southeast of Shanghai over the East China Sea, see Figure 1) from
October 2011 to August 2012 (Wang F. W. et al., 2015) and at Okinawa island (the same place as
Hedo in Figure 1) in the western Pacific Ocean from October 2009 to October 2010 (Kunwar and
Kawamura, 2014) were collected and used in this study. BC observations were conducted at Fukue
Island of western Japan using a continuous soot-monitoring system (COSMOS) (Figure 1) by Kanaya
et al. (2016) from 2009 to 2015.
Ground observations of AOD were obtained from the Aerosol Robotic Network (AERONET,
https://aeronet.gsfc.nasa.gov/, last access: 2020/06/03). Level 2 AOD observations for the year 2014
were collected at 7 coastal sites shown in Figure 1. Hourly and monthly mean observations were
derived from raw data and used for model comparison and statistics calculation. AOD at 550 nm was
used to match the model output. The level-3, daily deep blue global AOD product (in $1° \times 1°$ horizontal
resolution and at 550nm) retrieved by VIIRS sensor onboard the SNPP satellite platform (Sayer et al.
2018) were also collected to examine AOD spatial distribution.

3 Model validations
In this section, the model results for OC, BC, $PM_{10}$, $PM_{2.5}$, sodium concentrations, and AOD were
compared with a variety of observations from research cruise and monitoring networks to help
evaluate the model ability over wide areas from eastern China to the western Pacific Ocean. Because
the above comparison was for total OC mass concentration, we also compared the simulated SOA
from marine sources to cruise measured SOA tracer to examine the model performance for marine
organic aerosols.

3.1 Particulate matters ($PM_{10}$ and $PM_{2.5}$), sodium ($Na^+$) and gas precursors
As particulate matter in remote marine atmosphere is mainly composed of sea salt, the model



performance for $PM_{10}$ and $PM_{2.5}$ may partly reflect the model ability for sea salt simulation, which is
crucial to the estimation of MPOA emission.

Because the focus of this study is seasonal variation, the hourly $PM_{10}$ and $PM_{2.5}$ observations and

corresponding simulations were averaged to be monthly means and shown in Figure 2. In general,
RIEMS-Chem performed quite well in simulating monthly variation of $PM_{10}$ concentrations at both
the EANET sites (Figure 2a~2i) and CNEMC sites (Figure 2j~2l) for the year 2014, although model
biases still occurred at some sites, such as the underprediction in winter and spring in Jeju (Figure 2g)
and Imsil (Figure 2i) and the overprediction in May in Oki (Figure 2d) and Rishiri (Figure 2a). It was
striking that $PM_{10}$ concentration peaked in May and was lowest in August at all Korean sites and
northern Japanese sites over northeast Asia (Figure 2a~2d and 2g~2i), which could be attributed to the
long-range transport of mineral dust from north China and Mongolia in spring and to the
southwesterlies consisting of mainly marine air masses in summer. It was noteworthy that the model
simulated seasonality and magnitude of $PM_{10}$ agreed quite well with observations at the four island
sites of northern Japan (Rishiri, Tappi, Sado, and Oki) (Figure 2a~2d), where sea salt aerosol played a
more important role than those sites in Korea, implying sea salt concentrations could also be well
reproduced by the model. The seasonality of $PM_{10}$ concentration at Hedo (Figure 2e) was different
from above, showing high values in winter as well besides the peaks in spring, which indicated
potential influence of continental anthropogenic sources under prevailing northwesterlies. The $PM_{10}$
level at Ogasawara (Figure 2f) was much lower than those at the other sites and its seasonality was
characterized by the minimum in summer (5 μg m$^{-3}$) and the maximum in spring. The model
reasonably reproduced the seasonality at Hedo (Figure 2e) and Ogasawara (Figure 2f) as well,
although it generally predicted lower values at Hedo and higher values at Ogasawara. As for $PM_{10}$
concentrations at the CNEMC sites of eastern China, the model simulated $PM_{10}$ concentrations very
well for Shanghai (Figure 2k) and Fuzhou (Figure 2l) in terms of both monthly variation and
magnitude, showing higher values in spring and the maximum in winter in Shanghai, and an almost
stable level around 60 μg m$^{-3}$ in Fuzhou throughout the year except for the elevated value in January.
The $PM_{10}$ level in Qingdao (Figure 2j) was higher than those in Shanghai and Fuzhou, and reached the
maximum of 170 μg m$^{-3}$ in January due to anthropogenic sources and the peak in March was resulted
from the effect of mineral dust.

The monthly variations of $PM_{2.5}$ concentrations at Rishiri, Sado, and Oki (Figure 2m~2o) were



similar to those of PM$_{10}$, but the peaks in May were not as evident as those of PM$_{10}$, because mineral
dust comprises a small fraction of fine particles and has less effect on PM$_{2.5}$ variation. The model
reproduced PM$_{2.5}$ concentrations very well at the three coastal sites of eastern China (Figure 2p~2r)
and the monthly variation of PM$_{2.5}$ concentrations resembled those of PM$_{10}$, because fine particle
accounts for a large fraction of PM mass in these Chinese megacities due to the dominant effect of
anthropogenic sources.
Table 1 shows that for all the 9 EANET sites, the overall mean PM$_{10}$ concentration was 30.0 μg
m$^{-3}$ from observation and 28.5 μg m$^{-3}$ from simulation, with the overall Pearson correlation coefficient
(R) of 0.65 (0.48~0.64) and the normalized mean bias (NMB) of -5% (-27~36%). For PM$_{2.5}$, the mean
concentrations averaged over the EANET sites were 10.9 μg m$^{-3}$ from observation and 12.3 μg m$^{-3}$
from simulation, with R and NMB of 0.61 (0.53~0.64) and 12% (0~21%), respectively. The annual
mean observed and simulated PM$_{10}$ concentrations at the 3 CNEMC sites (Table 2) were 81.6 μg m$^{-3}$
and 80.7 μg m$^{-3}$, with R and NMBs of 0.65 (0.38~0.61) and -1% (-4~1%), respectively, while the
annual mean observed and simulated PM$_{2.5}$ concentrations, R, and NMB were 46.6 μg m$^{-3}$, 43.4 μg
m$^{-3}$, 0.70 (0.44~0.72), and -7% (-12~0%), respectively. The good performance statistics shown in
Table 1 and Table 2 suggest a good skill of RIEMS-Chem in reproducing PM levels from the coastal
regions of east China to the remote western Pacific. Figure 2, Table 1, and Table 2 also illustrate that
the spatial distribution of PM exhibited higher concentrations at the continental (costal) sites
(CNEMC sites, Jeju, Kanghwa, and Imsil) and lower concentrations at the remote island site
(Ogasawara) over the western Pacific, which were also reasonably reproduced by RIEMS-Chem.
Seasonal mean statistics of PM$_{10}$ and PM$_{2.5}$ concentrations at the EANET and CNEMC sites were
also listed in Table 1 and Table 2. Statistics for spring (March-April-May, MAM), summer
(June-July-August, JJA), autumn (September-October-November, SON), and winter
(December-January-February, DJF) were calculated. PM$_{10}$ observations generally exhibited higher
concentrations in MAM and DJF, moderate concentrations in SON, and lower concentrations in JJA at
most sites covering coastal areas (CNEMC sites, Jeju, Kanghwa, and Imsil) and remote islands (e.g.
Oki, Hedo, and Ogasawara). The model reproduced such seasonal variation of PM$_{10}$ reasonably well
although some underestimations occurred from winter to spring at Jeju and Imsil (Figure 2g, 2i),
which could be attributed to the uncertainties in emissions (anthropogenic, biomass burning).
Comparison with observations of Sodium (Na$^+$) concentration at 6 Japan coastal/island sites from





EANET is conducted to further examine the model performance for sea salt. The modeled sodium is
estimated to be 38.56% of sea salt mass (Kelly et al., 2010), and the agreement between observation
and model simulation is generally satisfactory at all sites except at Oki in December, when the model
largely underpredict $Na^+$. The model well reproduces the seasonality of sodium concentration, with
the maximum in winter and the minimum in summer (Figure 3). The model predicts sodium
concentration best at Ogasawara, with the correlation coefficient of 0.85 and NMB of 5%. The overall
correlation coefficient for all sites is 0.50, with NMB of -11% (Table S1).
In all, RIEMS-Chem was able to reasonably reproduce the spatial distribution and seasonal
variation of $PM_{10}$, $PM_{2.5}$, and sodium concentrations in the marine environment of the western Pacific.
The above good performances give us confidence in the estimation of marine sea salt emission.
In addition, the overall statistics were generally acceptable for gas precursors ($O_3$, $SO_2$, and
$NO_x/NO_2$), indicating atmospheric chemistry processes could be reasonably represented by the model
over the western Pacific. (see statistics in Table S2)

3.2 Carbonaceous aerosols
Modeled BC and OC concentrations were compared with observations from research cruises and
from previous publications at coastal/remote islands. BC is considered to be inert and chemical
inactive, so it is governed solely by physical processes and a good indicator of long-range transport.
The analysis of BC can help identify regions with large continental influence.

3.2.1 Comparison with research cruise measurements
Figure 4a shows the observed and simulated daily BC concentrations along the cruise track
during the spring campaign. An obvious spatial gradient was found for BC concentration, which was
characterized by apparent higher concentrations of 0.5~4.2 µg m$^{-3}$ over the marginal seas of China
(the Yellow Sea and East China Sea, 18~19 March and 21~22 April) and very low concentrations of
~0 to <0.2 µg m$^{-3}$ over open oceans (during most of the measurement days). It is interesting to note
that an observed BC peak occurred on 21 March, which could be attributed to the long-range transport
of biomass burning plumes from northeast Asia (Luo et al., 2016; 2018). The model generally
reproduced the spatial and temporal variations of BC concentration during the campaign period;
however, the BC peak on 21 March was missed by the model simulation. Uncertainties in biomass





burning emission could be responsible for such model bias. On average, the measured and simulated
BC concentrations during this campaign onboard the Dongfanghong II cruise were 0.49 μg m$^{-3}$ and
0.55 μg m$^{-3}$, respectively, with the R and NMB of 0.87 and 13% (Table 3).
Figure 4b shows the daily mean OC concentrations from observation and model simulation for
the same cruise. In general, the observed OC exhibited a similar spatial distribution and temporal
variation to that of BC, with higher concentrations over the marginal seas and relatively lower
concentrations over open oceans. The model generally captured the spatial-temporal features along the
cruise track. Like BC, the observed OC concentrations were high on 21 and 25~26 March mainly due
to the continental outflow of biomass burning emissions from northeast Asia, and the model largely
underpredict the high OC observation in these days. It is noteworthy that two OC peaks appeared on
10 and 12 April when the ship was over the open ocean east of Japan (the ship location was around
33.5°N, 146.0°E on 10 April and around 36.5°N, 145.0°E on 12 April, approximately 400~500 km to
the east of Japan), whereas the elevation of BC concentration was not evident. Because BC and OC
are often originated from the same anthropogenic and biomass sources, the inconsistency in daily
variation between BC and OC in these areas implied a potential influence of marine sources rather
than that from anthropogenic and biomass burning emissions. Coincidentally, during these days, daily
Chl-a concentrations over the oceanic areas east of Japan (the region of 35°N to 43°N and 140.0°E to
150.0°E, north to the ship location) reached as high as 45 mg m$^{-3}$, as a comparison, the monthly mean
Chl-a concentration in April over the same region was in a range of 2 to 14 mg m$^{-3}$. The apparent
higher Chl-a concentration during these days could induce changes in marine primary organic
emissions. On 10 April, the wind direction in the vicinity of the cruise was mainly southwesterly,
backward trajectory (figure not shown) indicates that air parcels travelled over low Chl-A regions to
the southwest of the cruise, implying a small effect of MOA. On this day, the fraction of land-OC (OC
originated from continental sources) in total OC was 68%, which was larger than that of marine
organic carbon (marine-OC) (32%) as shown in Figure 4b. On 12 April, northwesterly winds prevailed
over the cruise region, with backward air trajectory traveling over strong Chl-A regions to the
northeast of Japan (figure not shown), marine-OC aerosols produced from the bloom regions could be
blown to the southeast where ship located, leading to the elevation of OC concentrations (Figure 4b).
Marine-OC (percentage contribution of 74%) dominated over land-OC (26%) in the total OC
concentration on this day. The model improves OC simulation on 10 and 12 April when considering





marine organic aerosols (marine-OC in Figure 4b). The cruise campaign average OC concentration
was 1.20 μg m$^{-3}$ from observation and 1.14 μg m$^{-3}$ from simulation, with the R and NMB of 0.66 and
-5%, respectively (Table 3). The inclusion of marine-OC (including both primary and secondary OC)
reduced the model bias from -33% to -5% along the cruise. The average contribution of marine-OC to
the total OC mass in the marine atmosphere was approximately 29% along the cruise, with lower
contributions of 11~27% over the marginal seas of China (18~19 March and 21~22 April) and higher
contributions of 32~74% over the open oceans (5~18 April) (Figure 4b), demonstrating an increasing
importance of marine organic aerosols to total OC mass from the marginal seas to remote open
oceans.
Shown in Figure 4c is OC samples collected onboard the KEXUE-1 Research Vessel over the
East China Sea during the early summer campaign and the corresponding model results along the
cruise track. There were four OC peaks observed during the campaign, with three occurring over the
northern parts of the East China Sea (on 20 May, 26~29 May, and 1~5 June) and one over the southern
part of the East China Sea on 22 May. The model reproduced the OC variation quite well during most
of the cruise track, capturing the three OC peaks over the northern parts of the East China Sea
although low biases occurred for the first peak (over the area of 27.5°N to 30.0°N and 121.6°E to
121.9°E). The model missed the second OC peak on 22 May over the southern part of the East China
Sea (over the area of 22°N to 23°N and 121.5°E to 122.2°E). Kang et al. (2018) proposed that this
peak was seriously affected by biogenic and biomass burning emissions from Southeast Asia
(Philippines) because the OC concentrations from 21 to 25 May were characterized by high
abundance of sesquiterpene-derived SOA which was mainly originated from terrestrial photosynthetic
vegetation (e.g. trees and plants). Uncertainties in emission inventories, such as missing some
biogenic sources (e.g. fungal spores, Fröhlich-Nowoisky et al. 2016) could be partly responsible to the
model biases. In addition, some regions of Southeast Asia (e.g. Philippines) were not included in the
study domain, instead, their influence on the study domain was represented by chemical boundary
conditions from MOZART simulation, so, the uncertainties in chemical boundary conditions may also
contributed to such biases. At the time of the third (25°N to 26°N and 118.8°E to 121.7°E) and fourth
(28°N to 28.7°N and 119.6°E to 122.7°E) OC peaks, the ship was close to the shore and
predominately affected by continental sources (such as anthropogenic and biomass burning emissions),
the model captured the peaks quite well in terms of both temporal variation and magnitude. On



average, the observed and simulated OC concentrations from the KEXUE-1 cruise were 4.26 μg m$^{-3}$
and 3.68 μg m$^{-3}$, respectively, with R and NMB of 0.75 and -13% (Table 3). The inclusion of
marine-OC reduced the NMB from -19% to -13%. Along the cruise track, marine-OC was estimated
to account for 6% (1~60%) of the total OC mass on average, with lower contribution over the seas
close to the continent (1~9%) and higher contribution over the seas far from the continent (7~60%).
During the KEXUE-1 cruise campaign, the contribution of marine-OC to total OC mass was
obviously lower than that during the spring campaign conducted by the Dongfanghong II, because this
cruise over the marginal seas of China was more affected by continental outflow of anthropogenic and
biomass emissions compared with that mainly over the open oceans.

3.2.2 Comparison with measurements at island and coastal sites
Figure S1 shows the modeled BC is generally consistent with observations at island sites
(Huaniao, Fukue, Okinawa, Chichijima) in terms of both spatial distribution and seasonal variation,
indicating a good skill of RIEMS-Chem in representing the physical processes and long-rang transport
of carbonaceous aerosols over the western Pacific.
OC observations are limited in the western Pacific Ocean. We collected observations at islands
from previous publications (Boreddy et al., 2018; Kunwar and Kawamura, 2014; Wang F. W. et al.,
2015) for model comparison. Figure 5 shows the model simulated and observed seasonal/monthly
mean OC concentrations at the three islands. It should be kept in mind that the observations are
averages of different years. At Huaniao Island (Figure 5a), a distinct seasonality of OC observation
was shown, with the highest OC concentration of 4.7 μg m$^{-3}$ in DJF, followed by 3.7 μg m$^{-3}$ in MAM
and 3.8 μg m$^{-3}$ in SON, and the minimum of 1.1 μg m$^{-3}$ in JJA (Table 4). It was encouraging that
RIEMS-Chem reproduced the OC seasonality at Huaniao Island quite well (Figure 5a), despite the
different years between simulation and observation. The simulated OC was also divided into land-OC
and marine-OC to quantify the relative contribution of these sources to total OC mass. The simulated
annual mean OC concentration was 3.2 μg m$^{-3}$, in which 2.6 μg m$^{-3}$ (81%) was contributed by
land-OC and 0.6 μg m$^{-3}$ (19%) by marine-OC (Table 4). The simulation was very close to the
observation of 3.3 μg m$^{-3}$ (Table 4). It was striking that the inclusion of marine-OC obviously
improved the model performance, reducing the NMB from -21% to -3%, although the improvement of
prediction for SOA from land source may also reduce the model bias at Huaniao island. It was



noteworthy that marine-OC exhibited the maximum value in MAM and the minimum value in JJA.
The higher Chl-a concentration over the East China Sea in MAM might be responsible for the
maximum at Huaniao Island (Figure 7h and Table 7), whereas the lowest sea salt emission flux could
result in the minimum in summer (Table 7). In terms of seasonal mean, marine-OC accounted for 12%,
22%, 19%, and 23% of the total OC concentration in DJF, MAM, JJA, and SON, respectively, with an
annual mean contribution of 19% at Huaniao Island. The lowest relative contribution (12%) of
marine-OC in winter was attributed to the maximum anthropogenic OC emissions in eastern China in
this season.
At Okinawa (Figure 5b), the observed total OC showed the maximum in MAM, followed by that
in JJA, and the lower ones in DJF and SON during October 2009-2010. Figures 5a and 5b also show
that the seasonal cycling of OC concentration at Okinawa (Figure 5b) differed a lot from that at
Huaniao Island (Figure 5a). The high OC concentration in summer at Okinawa could be attributed to
higher SOA produced by local biogenic VOC emissions (Kunwar and Kawamura, 2014). The model
generally reproduced the seasonal variation of OC except that it predicted lower OC level in summer,
which could be due to the exclusion of local biogenic VOC emissions in the CAMS-GLOB-BIO
emission inventory. In terms of annual average, the observed OC concentration was 1.8 μg m$^{-3}$, larger
than the simulations of 1.3 μg m$^{-3}$ from the FULL case including marine-OC and of 1.1 μg m$^{-3}$ from
the case excluding marine organic emissions (Table 4). The inclusion of marine organic emissions
improved OC simulation at Okinawa, reducing the NMB from -39% to -28%. It was estimated that
marine-OC accounted for 18%, 17%, 10%, and 18% of total OC mass concentration at Okinawa in
DJF, MAM, JJA, and SON, respectively, with an annual mean contribution of 17%. The relatively
smaller contribution of marine-OC to the total OC mass at Okinawa than that at Huaniao Island (19%)
could be attributed to the higher Chl-a concentration and MPOA emission flux in the marginal seas of
China than those over remote western Pacific south of Japan (Figure 7).
Long-term average (2001-2012) of monthly mean OC concentrations at Chichijima Island
reported by Boreddy et al. (2018) and the simulated monthly mean OC concentration in 2014 were
shown in Figure 5c. The observations show higher OC levels from January to March mainly due to
continental outflows. It was noticed that the simulated OC levels in April-May were apparently higher
than observations, which could be associated with different time periods between observation and
simulation, and with potentially stronger continental outflows and bloom in spring 2014 than those of



ten-year averages. OC observations were relatively lower in summer and autumn due to the
dominance of high-pressure system and pristine ocean air mass over the western Pacific (Figure 9d
and 9e). The model tended to predict lower OC level in summer and autumn (Figure 5c). Boreddy et
al. (2018) indicated that in summer and autumn, OC at Chichijima was often influenced by long-range
transport of biomass burning plumes from Southeast Asia, which was not well represented in the
model (using chemical boundary conditions from MOZART-4 instead) and led to low model bias. On
average, the annual mean OC concentration was 0.76 $\mu$g m$^{-3}$ from observation, and 0.78 $\mu$g m$^{-3}$ from
the FULL case, and 0.65 $\mu$g m$^{-3}$ without considering marine-OC (Table 5). The inclusion of marine
organic emissions reduced the annual mean NMB from -13% to 3% and enhanced the correlation
coefficient from 0.56 to 0.6 at this site. The apparent better simulation from the FULL case indicated
the necessity of inclusion of marine organic emissions for simulating OC over the remote oceans of
the western Pacific. Both observation and model simulation revealed higher seasonal mean OC
concentrations in MAM (observed: 0.83 $\mu$g m$^{-3}$, simulated: 0.91 $\mu$g m$^{-3}$) and DJF (observed: 0.90 $\mu$g
m$^{-3}$, simulated: 1.2 $\mu$g m$^{-3}$) when the measurement site was frequently influenced by continental
outflows, whereas lower concentrations in JJA (observed: 0.65 $\mu$g m$^{-3}$, simulated: 0.47 $\mu$g m$^{-3}$) and
SON (observed: 0.66 $\mu$g m$^{-3}$, simulated: 0.57 $\mu$g m$^{-3}$) when clean maritime air masses or biomass
burning plumes from Southeast Asia (e.g. Philippine) influenced this region. The highest marine-OC
concentration was 0.19 $\mu$g m$^{-3}$ in MAM, followed by 0.16 $\mu$g m$^{-3}$ in DJF and 0.11 $\mu$g m$^{-3}$ in SON, and
the lowest one of 0.05 $\mu$g m$^{-3}$ in JJA. However, the percentage contribution of marine-OC to the total
OC mass was estimated to be largest in SON (20%), followed by 18% in DJF, 16% in MAM, and
lowest in JJA (10%), with an annual mean contribution of 16% (Table 5). The largest contribution in
SON was associated with the relatively lower total OC concentration as shown in Figure 5c. The
relative contribution from marine-OC to total OC at Chichijima Island resembled that at Okinawa in
terms of annual and season averages.

The above comparison against a variety of OC observations demonstrated a generally good skill

of RIEMS-Chem in simulating OC over the western Pacific in terms of seasonal variation and
magnitude. The model results from the FULL case indicated that including marine organic emissions
improved OC simulation over the western Pacific Ocean.

3.2.3 SOA over the western Pacific



Recently, Guo et al. (2020) reported SOA observations in the marine atmosphere from the
marginal seas of east China to the northwest Pacific Ocean. The measurements were conducted on
three research cruises in the spring and early summer of 2014 and in the spring of 2017. Total
suspended particulate (TSP) samples were collected from 19 March to 21 April 2014 over the
northwestern Pacific Ocean (NWPO), from 30 April to 17 May 2014 over the Yellow and Bohai seas
(YBS), and from 29 March to 4 May 2017 over the South China Sea (SCS). SOA concentration was
derived by using a tracer-based method. The measured SOA concentrations were 467 ng m$^{-3}$ over the
YBS, 617 ng m$^{-3}$ over the SCS, and 155 ng m$^{-3}$ over the NWPO, respectively. The model simulated
period and regional mean SOA concentrations were 664 ng m$^{-3}$ over the YBS, 466 ng m$^{-3}$ over the
SCS, and 157 ng m$^{-3}$ over the NWPO, which were generally consistent with the above observations,
although the study periods are not exactly the same. Guo et al. (2020) also presents the tracer-based
estimations of isoprene and monoterpene derived SOA in the air masses from ocean (assuming marine
sources), which were 1.7 ng m$^{-3}$ and 0.3 ng m$^{-3}$, respectively, over the western Pacific to the southeast
of Japan, whereas the modeled SOA concentrations produced from marine isoprene and monoterpene
emissions along the cruise track were 1.55 ng m$^{-3}$ and 0.28 ng m$^{-3}$, respectively, generally agreeing
with the tracer-estimation. However, it should be mentioned that there could be uncertainties in such
comparison. First, the isoprene- and monoterpene-derived SOA tracers in the air masses categorized
as marine sources by Guo et al (2020) might include SOA tracers from terrestrial isoprene and
monoterpene under the prevailing northwesterly winds in spring, which could bias the estimation high;
second, the measured tracer could just comprise a part of total SOA tracers, which might bias the
estimation low. Despite these uncertainties, the cruise measured SOA concentration derived from
marine isoprene and monoterpene was approximately several ng m$^{-3}$ over the western Pacific, and it
can reach approximately 10 ng m$^{-3}$ even through dividing by a mass fraction of tracer compound to
yield the concentration of total SOA tracers. It was noteworthy that both observation and model
simulation exhibited a decreasing SOA concentration from marginal seas of China to remote oceanic
areas. In all, the model reproduced the SOA levels in the marine atmosphere of the western Pacific
Ocean reasonably well.
The comparison of the magnitudes between SOA and OA mass (1.4 times OC mass)
concentrations shown above indicates that SOA concentration was approximately 1~2 orders of
magnitude lower than OA over the western Pacific. Previous observation studies using the





tracer-based approach also indicated that the percentage contribution of SOA to OA was quite low
over some marine areas (Fu et al., 2011; Hu et al., 2013; Bikkina et al., 2014; Zhu et al., 2016). For
example, at Okinawa island, even considering all biogenic sources (including isoprene, monoterpene,
and sesquiterpene of both terrestrial and oceanic origins), the measured concentration of total
biogenic-SOA tracers was still less than 100 ng m$^{-3}$, with majority of SOA tracers from local terrestrial
biogenic emissions (Zhu et al., 2016). The above studies suggested that primary organic aerosols were
more important in remote marine atmosphere.

3.3 Aerosol optical depth

Figure 6 shows the temporal variations of the observed and simulated monthly mean AOD at the

7 AERONET sites. In general, RIEMS-Chem simulated the monthly mean AOD reasonably well in
terms of magnitude and monthly variation at almost all sites, although some biases occurred during
some months, such as the overpredictions in August at Fukuoka and in April at EPA-NCU, and the
underprediction in July at Yonsei University. For the sites in the northern oceanic areas (Ussuriysk,
Yonsei_University, Gwangju_GIST, and Fukuoka, Figure 6a~6d), both observations and simulations
generally exhibited higher AOD values in summer (JJA), moderately high AOD values from late
winter (JF) to spring (MAM), and relatively lower AOD values in autumn (SON). The simulated
higher inorganic aerosol concentrations in summer and late spring months could be responsible for the
higher AOD values in these regions. Besides, the higher relative humidity in summer due to the
predominant influence of maritime air masses also contributed to the maximum AOD values during
summer months (JJA) at these sites. On the other hand, for the sites in the southern oceanic areas
(EPA-NCU and Chen-Kung_Univ, Figure 6e and 6f), the monthly mean AOD was apparently higher
from March to April and remained low levels during the rest months. The above AOD peaks in spring
could be attributed to the continental outflows of biomass burning plumes originated from Southeast
Asia, which were most active in springtime in those regions (Hsiao et al., 2017; Tao et al., 2020).
Table 6 shows the performance statistics for hourly AOD at these AERONET sites. The overall annual
mean AOD for the 7 sites was 0.34 from model simulation, which was very close to the observation of
0.36, with the NMB of -6% and the overall correlation coefficient of 0.54 (0.40~0.67). The statistics
indicate that the model was able to reproduce aerosol optical properties over the coastal regions and
islands around the western Pacific Ocean.



The model simulated annual mean AOD at 550 nm are also compared with the VIIRS retrievals
(Figure S2), which indicates the model is generally capable of reproducing AOD distribution and
magnitude in the study domain. The generally high model bias over the western Pacific could be
attributed to potential overpredictions of inorganic aerosol concentration and relative humidity. AOD
reflects the column integrated extinction coefficient due to all aerosols.
At the AERONET sites, the model simulated annual mean percentage contribution of MOA to
AOD varied from 1.4% to 3.2% with an overall average of 1.9%. For the oceanic VIIRS region, the
mean contribution of MOA to AOD was approximately 2%.

4 Model results
4.1 Marine primary organic and isoprene emissions
Figure 7 shows the estimated annual and seasonal mean MPOA emission rates over the western
Pacific of East Asia. The MPOA emission mainly occurred over two hotspot regions: the marginal
seas of China including the East China Sea, the Yellow Sea, and the Bohai Sea (EYB, denoted in
Figure 7a) and the northern parts of the western Pacific northeast of Japan (NWP, denoted in Figure
7a), with annual mean emission rates varying from $0.9\times10^{-2}$ $\mu g$ $m^{-2}$ $s^{-1}$ to $1.8\times10^{-2}$ $\mu g$ $m^{-2}$ $s^{-1}$. In SON,
high MPOA emission occurred in both the EYB and NWP regions, with the maximum up to $3.5\times10^{-2}$
$\mu g$ $m^{-2}$ $s^{-1}$ in the NWP (Figure 7e), whereas MPOA emission was very low over the EYB in JJA
(Figure 7d). The maximum seasonal mean emission rate of MPOA approached $3.6\times10^{-2}$ $\mu g$ $m^{-2}$ $s^{-1}$
over the Yellow Sea in DJF (Figure 7b), which was approximately 1/10 of the annual mean
anthropogenic POA emission rate in north China (on the order of $1.0\sim3.0\times10^{-1}$ $\mu g$ $m^{-2}$ $s^{-1}$). Table 7
presents the seasonal and annual averages of MPOA emission averaged over the western Pacific and
the EYB and NWP regions. In terms of oceanic average of the western Pacific, the mean MPOA
emission generally exhibited the largest emission rate in SON ($0.20\times10^{-2}$ $\mu g$ $m^{-3}$ $s^{-1}$), moderately high
emission rates in DJF ($0.18\times10^{-2}$ $\mu g$ $m^{-2}$ $s^{-1}$) and MAM ($0.17\times10^{-2}$ $\mu g$ $m^{-2}$ $s^{-1}$), and the lowest one in
JJA ($0.08\times10^{-2}$ $\mu g$ $m^{-2}$ $s^{-1}$), with an annual average of $0.16\times10^{-2}$ $\mu g$ $m^{-2}$ $s^{-1}$ (Table 7). It is interesting to
note that the seasonal variation of MPOA emission was not consistent with that of Chl-a concentration,
which exhibited higher values in SON and JJA and the lowest one in DJF (Table 7). This is because
MPOA emission rate is determined by the combined effect of Chl-a concentration and sea salt
emission flux, and sea salt flux is mainly controlled by surface wind speed according to the scheme of

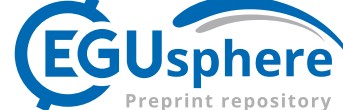



Gong (2003). In terms of seasonal and domain average over the western Pacific, the maximum Chl-a
concentration and the second largest sea salt emission flux in SON led to the largest MPOA emission
in autumn (Table 7). However, although Chl-a concentration was also high in JJA (1.07 mg m$^{-3}$, Table
7), the sea salt flux was minimum in JJA (0.14 µg m$^{-2}$ s$^{-1}$, Table 7) due to the weakest wind speed (3.0
m s$^{-1}$, Table 9), resulting in the lowest MPOA emission in summer (Table 7). Although the sea salt
emission flux reached the maximum in DJF (Table 7) due to the largest wind speed in this season
(Table 9), the winter Chl-a concentration was lowest, leading to a moderate MPOA emission in winter
(Table 7), in a similar magnitude to that in spring when moderately high Chl-a concentration and
relatively low sea salt flux occurred. In all, the MPOA emission rate over the western Pacific exhibited
an apparent seasonality of SON > DJF ≈ MAM > JJA.
For the EYB region, the maximum MPOA emission occurred in winter (DJF) (Figure 7b and
Table 7) with a seasonal and domain average of 1.2×10$^{-2}$ µg m$^{-2}$ s$^{-1}$, which was 10 times larger than
the minimum of 0.12×10$^{-2}$ µg m$^{-2}$ s$^{-1}$ in summer (JJA) (Figure 7d and Table 7). Although Chl-a
concentrations were similar between DJF and JJA, the sea salt flux in DJF was approximately 9 times
that in JJA (Table 7). So, the seasonality of MPOA emission in the EYB region was mainly
determined by that of sea salt emission flux due to the weak seasonal variation of Chl-a concentration.
Differently, in the NWP region, MPOA emission exhibited the maximum value in SON, followed by
those in MAM and DJF, and the lowest ones in JJA (Table 7). It is interesting to note that although
both the Chl-a concentration and sea salt emission flux were slightly higher in MAM than those in
SON, the MPOA emission (related to both Chl-a concentration and sea salt emission) was higher in
SON, which could due to the slight negative correlation between and sea salt emission in MAM, but
the slight positive one in SON. The MPOA emissions in winter and summer were in a similar level in
the NWP region, about 40% lower than that in autumn.
The distribution pattern of MPOA emission in the western Pacific from this study is similar to
those from previous model studies (Spracklen et al., 2008; Gantt et al., 2009; Huang et al., 2018), but
the magnitude of the simulated MPOA emission flux is larger than previous estimates. For example,
the annual mean MPOA emission rates over the western Pacific were estimated to vary from 0.1 to
approximately 12 ng m$^{-2}$ s$^{-1}$ in previous studies (Spracklen et al., 2008; Vignati et al., 2010; Gantt et
al., 2011; Long et al., 2011; Huang et al., 2018), whereas the estimates in this study ranged from 3 to
18 ng m$^{-2}$ s$^{-1}$ (Figure 7a). The larger marine POA emission estimated in this study could be attributed



to the application of the daily mean Chl-a concentration from satellite retrievals and of a finer model
grid resolution (60 km) compared with those in global models. On average, the annual MPOA
emission was estimated to be 0.78 Tg yr$^{-1}$ over the western Pacific (with an ocean area of $1.58\times10^7$
km$^2$) from this study. The regions of EYB and NWP comprised approximately 2% and 18% of the
western Pacific in terms of area, respectively, but they contributed 8% and 46% of the MPOA annual
emission (Tg y$^{-1}$). This study revealed that the EYB and NWP are important bloom regions,
accounting for more than half of the total MPOA emission over the western Pacific.

Table S3 presents the simulated marine isoprene emission fluxes in comparison with

observation-based estimates over the western Pacific of East Asia and other oceans from previous
studies. Over the western North Pacific, the observed marine isoprene emission flux showed larger
values in May (140~143.8 nmol m$^{-2}$ day$^{-1}$), a moderate value in August (~55.6 nmol m$^{-2}$ day$^{-1}$), and
the lowest one in winter (~21.4 nmol m$^{-2}$ day$^{-1}$). The model simulation generally agreed with
observation in terms of both seasonality and magnitude except for the low bias in spring (85~89 nmol
m$^{-2}$ day$^{-1}$ in spring, ~63 nmol m$^{-2}$ day$^{-1}$ in summer, and ~26 nmol m$^{-2}$ day$^{-1}$ in winter), which could be
associated with the different years. According to equations (2) and (3), both Chl-a concentration and
incoming solar radiation determine marine biogenic VOCs emission, the larger isoprene flux in May
was mainly due to the maximum Chl-a concentration in spring over the NWP region (Table 7). Over
the marginal seas of China, Li J. L. et al. (2017; 2018) observed higher marine isoprene emission flux
in July-August (~161.5 nmol m$^{-2}$ day$^{-1}$) than those in October-November (~48.3 nmol m$^{-2}$ day$^{-1}$) and
May-June (~36.1 nmol m$^{-2}$ day$^{-1}$) during 2013-2014. The model well reproduced the seasonal trend
and magnitude of isoprene flux, with corresponding mean values of 130 nmol m$^{-2}$ day$^{-1}$, 48 nmol m$^{-2}$
day$^{-1}$, and 35 nmol m$^{-2}$ day$^{-1}$ during the same periods of 2014, respectively. The apparently higher
isoprene flux in July-August was mainly resulted from the strongest solar radiation in summer,
although the Chl-a concentration was not highest in this season in the EYB region (Table 7). Table S3
also lists previously observed marine isoprene emission fluxes over the Southern Ocean and Arctic
Ocean in summer for reference. The domain-wide annual marine isoprene emission estimated over the
western Pacific was 0.015 Tg yr$^{-1}$ in this study. Arnold et al. (2009) calculated with GEOS-Chem
model a global-annual isoprene emission of 0.31 Tg yr$^{-1}$. However, some previous studies (Arnold et
al., 2009; Booge et al., 2016) found the emission flux calculated by current marine isoprene emission
schemes tended to yield lower isoprene concentration in marine atmospheres compared with



observations.

4.2 Marine organic aerosols and their relative importance

Annual and seasonal mean near surface MOA concentrations, MSOA concentrations, and the

percentage contributions of MOA to total OA mass in the study domain were shown in Figure 8. The
spatial distributions of MOA concentrations (Figure 8a~8e) generally resembled those of MPOA
emissions (Figure 7a~7e). It is remarkable that MPOA concentration (MOA minus MSOA) was
approximately 1~2 orders of magnitude higher than MSOA concentration (with concentration of
several ng m$^{-3}$) in the western Pacific (Figure 8a~8e vs Figure 8f~8j), indicating that MPOA
constituted a dominant fraction of MOA, which will be discussed below. Figure 8a shows that high
MOA concentrations mainly occurred over the EYB and NWP regions, with the annual and regional
averages being 0.48 μg m$^{-3}$ and 0.59 μg m$^{-3}$, respectively (Table 8), accounting for 13% (6~30%) and
42% (30~60%) of total OA mass in these two regions, respectively (Figure 8k and Table 8). The larger
MOA contribution over the NWP was attributed to the high MOA level and the relatively low total OA
level there. It is noticed that MOA even influenced the coastal areas of eastern China. The annual
mean MOA concentration decreased from approximately 0.5 μg m$^{-3}$ in coastal areas to 0.1 μg m$^{-3}$ in
the inland areas (Figure 8a), accounting for approximately 2% to 6% of the near-surface OA mass in
the coastal regions (Figure 8k). The maximum seasonal mean MOA concentration over the coastal
areas of eastern China could be up to 0.6 μg m$^{-3}$ to 0.8 μg m$^{-3}$ in MAM (Figure 8c) and SON (Figure
8e). The domain and seasonal mean MOA concentration over the western Pacific exhibited the
maximum value in MAM (0.37 μg m$^{-3}$), follow by that in SON (0.26 μg m$^{-3}$), and relatively lower
concentrations in JJA (0.23 μg m$^{-3}$) and DJF (0.21 μg m$^{-3}$) (Table 8). It was noteworthy that the
seasonality of MOA concentration was different from that of MPOA emission, which could be
attributed to the influence of different meteorological conditions and physical processes. In the
western Pacific, although MPOA emission peaked in SON (Table 7), MOA concentration peaked in
MAM (Table 8). It is noticed that precipitation was lowest and wind speed was low in MAM (Figure
9c and 9h, Table 9), leading to a smaller dry deposition velocity (Zhang et al. 2001) and the weakest
wet scavenging, both favored accumulation of MOA and thus resulted in the highest MOA level in
spring. On the contrary, due to the maximum wind speed and relatively more precipitation in DJF
(Figure 9b and 9g, Table 9), the mean MOA concentration was lowest in winter.





For the EYB region, northwesterly winds prevailed In DJF and SON and turned to northeasterly
winds over marginal seas of southeast China (Figure 9b and 9e), which transported MOA from the
major MPOA source region (EYB) to the northern part of the South China Sea (Figure 8b and 8e). As
wind speed over the EYB was low in MAM and JJA (Figure 9c and 9d, Table 9), MOA was mainly
restricted within this region (Figure 8c and 8d). In terms of seasonal average, MOA concentration
experienced its maximum in MAM, followed by those in DJF and SON, and the minimum in JJA
(Figure 8b~8e). The seasonal and regional mean MOA concentrations over the EYB were 0.62 μg m$^{-3}$,
0.54 μg m$^{-3}$, 0.52 μg m$^{-3}$, and 0.22 μg m$^{-3}$ for MAM, DJF, SON, and JJA, respectively (Table 8). The
different seasonality between MOA concentration (Table 8) and MPOA emission (Table 7) in the EYB
region could also be mainly attributed to meteorological conditions. The MPOA emission was
relatively low in MAM (Table 7), but the second lowest wind speed and less precipitation (Table 9)
favored aerosol accumulation, resulting in the highest MOA concentration in spring (Table 8). The
minimum MPOA emission and the maximum precipitation in JJA led to the minimum MOA
concentration in summer. Although MPOA emission was largest in SON and DJF (Table 7), the
maximum wind speeds (Table 9) led to stronger dry deposition of aerosols and thus a moderate MOA
concentration in the two seasons (Table 8).
MOA concentration over the NWP region exhibited apparent higher concentrations in MAM and
JJA than those in SON and DJF (Figure 8b~8e), with the regional and seasonal averages reaching 0.81
μg m$^{-3}$ in MAM, 0.80 μg m$^{-3}$ in JJA, 0.52 μg m$^{-3}$ in SON, and 0.23 μg m$^{-3}$ in DJF, respectively (Table
8). Using GEOS-Chem with a different marine organic aerosol emission scheme, Spracklen et al.
(2008) also showed that in the North Atlantic along the similar latitude bands to the NWP (~35°N to
~55°N), both observation and simulation exhibited higher OC concentrations in summer and spring
than in the other seasons at the Azores Island and Mace Head Island. The strong seasonality of MOA
over the NWP was also attributed to the combined effects of MPOA emission, wind speed, and
precipitation. In MAM, the high MOA concentration over the NWP was mainly due to the large
MPOA emission (Figure 7c and Table 7), which was just smaller than that in SON, and partly due to
the relatively weak dry deposition and wet scavenging caused by moderate wind speed and
precipitation in this season (Table 9). In JJA, although the MPOA emission was small, the lowest wind
speed and precipitation in JJA over the NWP (2.5 m s$^{-1}$ and 3.7 cm grid$^{-1}$ month$^{-1}$, Table 9) led to the
weakest dry deposition and wet scavenging of particles in summer, resulting in a long residence time





of MOA and consequently the high MOA concentration in summer over the NWP. In SON, although
the MPOA emission was largest over the NWP (Table 7), the mean wind speed was high over the
northern part of the NWP (Figure 9e) where MPOA emission mainly occurred (Figure 7e), leading to
strong dilution of MOA particles in autumn. Furthermore, the secondly largest precipitation over the
NWP in SON (Table 9) caused strong wet scavenging of particles, also contributed to the relatively
low MOA level. In DJF, the wind speed was largest, about 2 times those in the other seasons, and the
precipitation was also the maximum (Table 9, Figure 9b and 9g), leading to the lowest MOA
concentration in winter over the NWP (Figure 8b and Table 8).

As shown in Figures 8k~8o, MOA generally accounted for approximately 30% to over 60% of

total OA concentration over the remote oceans of high (>35°N) and low (<25°N) latitudes. The large
MOA/OA ratios over the remote oceans of high latitude (including NWP) could be attributed to the
high MOA concentration due to large marine emissions there; whereas, the large MOA/OA ratios over
the subtropical oceans of low latitude were mainly due to the low total OA level (small denominator).
Averaged over the NWP region, the annual mean MOA/OA ratio was 42%, with higher contributions
in MAM (52%) and SON (48%) and lower ones in DJF (36%) and JJA (32%) (Table 8). Although
MOA concentration over the NWP was secondly highest in JJA, its contribution was small because
OA transported from land sources also subject to weak dry deposition and wet scavenging, which led
to higher OA level and lower MOA/OA ratio. Over the EYB region, MOA accounted for
approximately 6% to 30% of the total OA in terms of annual mean (Figures 8k). In terms of annual
and regional average, the MOA/OA ratio was 13%, with higher ratios in SON (18%) and MAM (15%),
a moderate one in DJF (11%), and the lowest one in JJA (6%) (Table 8), similar to the seasonality over
the NWP. For the oceanic areas, the domain and annual mean MOA/OA ratio was 26%, indicating an
important contribution of MOA to airborne OA over the western Pacific. It was found that the
importance of MOA in total OA increased as the distance to the East Asian continent increased over
the western Pacific. It is also interesting to note that MOA even accounted for approximately 2~6% of
the annual mean OA mass over portions of southeast China (Figures 8k), and such contribution could
be as high as 8~10% in the coastal areas in SON (Figures 8o) and MAM (Figures 8m).

In all, both the MOA concentration and the MOA contribution to total OA were lowest in summer

(JJA) in the EYB region, which was mainly due to the much smaller MPOA emission in this season.
However, in the NWP region, although the MPOA emission was also lowest in summer, MOA



concentration in summer was in a same level as that in spring, and larger than those in the other seasons, because dry deposition velocity and precipitation were lowest in summer, which favored aerosol accumulation and a high level of MOA.

SOA produced by marine biogenic VOCs (isoprene and terpene) was on the order of $10^{-2}\sim10^{-3}$ μg $m^{-3}$ (Figure 8f~8j), which was much lower than the MPOA concentration. The spatial distribution of MSOA exhibited high concentrations over the EYB and NWP regions in terms of annual mean, with values up to 6 ng $m^{-3}$ (approximately 0.5% of MOA concentration) over these two regions (Figure 8f). MSOA concentration exhibited the maximum in JJA, with seasonal mean values of ~7 ng $m^{-3}$ to ~11 ng $m^{-3}$ extending from the marginal seas of China (EYB) to remote western North Pacific (NWP) (Figure 8i). MSOA distribution in MAM was similar to that in JJA but with lower mean concentrations (4~7 ng $m^{-3}$) over the EYB and NWP regions (Figure 8h). In SON (Figure 8j), MSOA concentrations were 2~4 ng $m^{-3}$ in the above two regions. In DJF (Figure 8g), MSOA concentration was lowest, with values of 0.4~2 ng $m^{-3}$ over the marginal seas of China and the southern parts of the western Pacific. The maximum seasonal mean MSOA concentration was up to 14 ng $m^{-3}$ over oceanic areas of the EYB to NWP regions in JJA, and the maximum daily mean MSOA value exceeded 28 ng $m^{-3}$ on some days, e.g. June 6~7 (figure not shown). Table 8 shows the domain and seasonal/annual averages of MSOA over the oceanic regions of concern. The annual mean MSOA concentrations were 2.2 ng $m^{-3}$, 4.1 ng $m^{-3}$ and 3.8 ng $m^{-3}$ averaged over the western Pacific, the EYB and NWP regions. It is striking that the domain average MSOA concentration consistently exhibited a distinct seasonality, with the maximum in summer and the minimum in winter throughout all the oceanic regions of the western Pacific, which was resulted from the combined effects of isoprene emission flux and meteorological conditions. The domain average MSOA concentrations reached the maximums of 3.9 ng $m^{-3}$, 7.5 ng $m^{-3}$, and 8.3 ng $m^{-3}$, respectively, over the western Pacific, the EYB and NWP regions in JJA. The seasonality of MSOA concentration over the western Pacific is similar to the simulation result from Myriokefalitakis et al. (2010). According to Table 8, the annual mean fraction of MSOA in MOA was estimated to be 0.8%, 0.9% and 0.6%, over the western Pacific, the EYB and NWP regions, respectively. The maximum and minimum fractions of MSOA in MOA averaged over the western Pacific occurred in JJA (1.7%) and DJF (0.3%), respectively, with the maximum regional and seasonal average MSOA fraction up to 3.4% in summer over the EYB region. Based on the GEOS-Chem model simulation, Arnold et al. (2009) indicated that SOA produced by marine isoprene





contributed only a very small fraction (0.01~1.4%) of the observed organic aerosol mass at remote
marine sites (Amsterdam Island in southern Indian Ocean, Azores and Mace Head islands in northern
Atlantic Ocean). In a global model simulation from Myriokefalitakis et al. (2010), the annual mean
marine isoprene and monoterpene derived SOA concentrations were approximately 0.4~1 ng m$^{-3}$
(accounting for ~0.4% of marine OA) over the western Pacific. Meskhidze et al. (2011) illustrated the
marine SOA from phytoplankton-derived isoprene and monoterpenes contributed <10% of surface
OM concentration of marine source in most areas of the western Pacific.

4.3 Direct radiative effect due to MOA

In this section, the direct radiative effect (DRE) due to MOA (DRE$_{MOA}$) over the western Pacific

of East Asia was estimated and analyzed. DRE is defined as the difference in net shortwave radiation
flux at TOA (or at the surface) induced by aerosols. The DRE of MOA is derived by the difference
between two radiation calls with all aerosols and with all aerosols but MOA (i.e., call two times in the
radiation module in the same simulation), reflecting an instantaneous change in shortwave radiation
fluxes induced by MOA. DRE of other aerosol components is estimated by using the same method.

Figures 10a to 10e show the annual and seasonal mean DRE$_{MOA}$ at TOA under all-sky condition.

MOA induced negative DRE over the entire western Pacific. Consistent with the spatial distribution of
MOA concentration, the maximum DRE$_{MOA}$ up to -0.9 W m$^{-2}$ occurred over the NWP region (Figure
10a). Over the EYB region, the other hotspot of MOA mass concentration, the DRE$_{MOA}$ was weaker,
with an annual mean DRE$_{MOA}$ of ~ -0.4 W m$^{-2}$ (Figure 10a). In terms of domain average, the annual
mean DRE$_{MOA}$ was estimated to be -0.27 W m$^{-2}$ over the western Pacific, smaller than that over the
NWP (-0.50 W m$^{-2}$) but similar to that over the EYB (-0.33 W m$^{-2}$) (Table 10). The weaker DRE$_{MOA}$
over the EYB than that over the NWP could be attributed to both the lower MOA concentration (Table
8) and lower relative humidity (73% vs 83%, Table 9). The mean DRE$_{MOA}$ over the western Pacific
was largest in spring (-0.38 W m$^{-2}$) and lowest in winter (-0.18 W m$^{-2}$) (Table 10), consistent with the
seasonality of MOA concentration.

In the NWP region, MOA induced the largest DRE up to -2.0 W m$^{-2}$ to the northeast of Japan in

MAM (Figure 10c) and followed by that in JJA (up to -1.5 W m$^{-2}$) (Figure 10d) mainly due to higher
MOA concentrations in the two seasons. The DRE$_{MOA}$ value was relatively low in SON (~ -0.7 W m$^{-2}$)
(Figure 10e), and it was lowest in DJF, with the maximum of just -0.4 m$^{-2}$ (Figure 10b) due to the



lowest MOA concentration in winter (Table 8). The regional and seasonal means of $DRE_{MOA}$ over the
NWP were estimated to be -0.76 W m$^{-2}$, -0.72 W m$^{-2}$, -0.32 W m$^{-2}$, and -0.21 W m$^{-2}$ in MAM, JJA,
SON, and DJF, respectively (Table 10). On the contrary, in the EYB region, $DRE_{MOA}$ exhibited a
different seasonal trend from that over the NWP, exhibiting the largest DRE in SON (Figure 10e),
moderate DREs in MAM (Figure 10c) and DJF (Figure 10b), and the lowest one in JJA (Figure 10d),
with corresponding mean values of -0.38 W m$^{-2}$, -0.34 W m$^{-2}$, -0.32 W m$^{-2}$, and -0.26 W m$^{-2}$,
respectively, for the four seasons (Table 10).

It is of interest to estimate the relative importance of MOA in directly perturbing solar radiation

compared with other types of aerosols over the western Pacific. Table 10 lists the simulated annual
and seasonal mean DREs due to sea salt and anthropogenic aerosols over the western Pacific and the
regions of NWP and EYB, respectively. It is noted that the annual mean $DRE_{MOA}$ was approximately
one third of that due to sea salt and one order of magnitude lower than that due to anthropogenic
aerosols on average over the western Pacific. Over the EYB region, $DRE_{MOA}$ was almost negligible
compared with that by anthropogenic aerosols due to the predominance of anthropogenic emissions
near the continent, however, it is noteworthy that $DRE_{MOA}$ was comparable in magnitude to the DRE
by sea salt, especially in springtime (-0.34 vs -0.36 W m$^{-2}$ in MAM) when MOA concentration
reached its maximum (Table 8). Over the remote oceans (NWP), $DRE_{MOA}$ was approximately 19% of
the DRE by anthropogenic aerosols due to weakened influence of continental anthropogenic emissions
over open oceans, and it was comparable in magnitude to the DRE by sea salt, especially in
summertime (0.72 vs 0.79 W m$^{-2}$) when sea salt emission flux was lowest (Table 7). The annual and
regional mean all-sky $DRE_{MOA}$ were approximately 10%, 5%, and 19% of the DREs due to
anthropogenic aerosols over the western Pacific, the EYB, and NWP, respectively, and it was
comparable in magnitude to the DREs by sea salt in the EYB and NWP regions.

It should be mentioned that due to the much smaller MSOA concentration than MPOA

concentration, the above $DRE_{MOA}$ was dominantly contributed by MPOA, similar to the findings from
previous studies (Arnold et al., 2009; Booge et al., 2016; Li et al., 2019). MPOA also dominated over
MSOA in the following IRE estimation.

4.4 Indirect radiative effect due to MOA

The first indirect effect refers to that aerosol particles tend to increase cloud droplet number





concentration, decrease cloud effective radius under stable cloud liquid water content, and thus
modify cloud optical properties and radiation, which is also denoted as the indirect radiative effect
(IRE) in this study. IRE due to marine organic aerosols ($IRE_{MOA}$) over the western Pacific of East Asia
was explored in this section. The first indirect radiative effect or IRE of MOA is estimated by the
instantaneous difference in net shortwave radiation flux at TOA (or at the surface) between two
radiation calls with all aerosols and with all aerosols but MOA in one simulation. The lower bound of
cloud droplet number concentration ($N_c$) is set to $10/cm^3$, which roughly represents $N_c$ in liquid clouds
in clean ocean conditions, according to satellite observations and global simulations. Hoose et al.
(2009) pointed out that the choice of low bound of $N_c$ may lead to uncertainties in IRE estimation. The
IRE calculation in this study can be written as follows:

$$IRE_{ai} = (F\downarrow - F\uparrow)_{all\ aerosols} - (F\downarrow - F\uparrow)_{all\ aerosols\ but\ ai}$$

Here $F\downarrow$ and $F\uparrow$ represents incoming and outgoing shortwave radiation fluxes, respectively, $a_i$ denotes
a specific aerosol component, e.g., MOA. Here, the calculation of IRE due to MOA (the sum of
MPOA and MSOA) is called the base case (BASE), from which a series of sensitivity simulations are
carried out below.

$N_c$ due to aerosol activation is diagnosed by the A-G scheme, then the cloud effective radius $r_e$ is
calculated as a function of $N_c$ and cloud liquid water content following the approach of Martin et al.
(1994), and the cloud optical parameters (liquid cloud extinction optical depth, single scatter albedo,
asymmetry factor etc.) are calculated by the scheme of Slingo et al. (1989), finally, shortwave
radiation fluxes are calculated by the CCM3 radiation scheme (Kiehl et al., 1996).

The annual and seasonal mean $IRE_{MOA}$ at TOA are shown in Figure 10f to 10j. $IRE_{MOA}$ was
negative in the entire domain, resulting from a series of changes in cloud properties induced by MOA,
i.e., an increase in cloud droplet number concentration, a decrease in cloud droplet effective radius, an
increase in cloud optical depth and cloud water path, and consequently more reflection of solar
radiation at TOA. The model simulated cloud properties have been compared against satellite
retrievals in spring 2014 in our previous study (Han et al., 2019), which indicated the model was able
to reasonably reproduce the major features of cloud distribution. It is remarkable that $IRE_{MOA}$ was
stronger than $DRE_{MOA}$ over the western Pacific, with the maximum annual mean of $IRE_{MOA}$ being
approximately three times the maximum of $DRE_{MOA}$, and the positions of their maximum values were
different. The annual mean $IRE_{MOA}$ of $-0.9 \sim -2.5$ W m$^{-2}$ distributed from southwest to northeast over



wide areas of the western Pacific (Figure 10f). It was evident that the strongest IRE$_{MOA}$ occurred in
spring (MAM), with the seasonal mean values of -1.2 ~ -3.0 W m$^{-2}$ over vast areas from the East
China Sea to the oceans east of Japan (Figure 10h). IRE$_{MOA}$ in SON was similar in distribution pattern
to that in MAM, with lower values in these regions (Figure 10j). The IRE$_{MOA}$ was weakest in JJA,
with the maximum up to -1.5 W m$^{-2}$ over a portion of the western Pacific east of Japan (Figure 10i),
whereas the IRE$_{MOA}$ value in DJF was between those in MAM and JJA with a similar distribution
pattern. The seasonal variation of IRE$_{MOA}$ was likely influenced by both the seasonal changes in cloud
amount and MOA concentration. In terms of domain average, the seasonal mean IRE$_{MOA}$ was
strongest (-0.94 W m$^{-2}$) in MAM over the western Pacific (Table 10), which was mainly attributed to
the maximum MOA concentration in spring (Table 8). IRE$_{MOA}$ was secondly strongest in SON (-0.7
W m$^{-2}$) because MOA concentration and cloud fraction (Figure S3j) were both high in autumn. The
weakest IRE$_{MOA}$ occurred in JJA, which was mainly attributed to both the lower MOA concentration
and cloud fraction in summer (Table 8, Figure S3i). Figure S4 further presents the monthly mean
distributions of Chl-a concentration, MPOA emission, MOA concentration, and IRE$_{MOA}$ in April,
when Chl-a concentration and MPOA emission resulting from phytoplankton were distinctly high in
the EYB and NWP regions (Figure S4a and S4b). It can be found that MOA was transported from the
high Chl-a regions to the southeast under northwesterly winds over the oceans (Figure S4c), resulting
in an elevated IRE$_{MOA}$ up to -5 W m$^{-2}$ over the western Pacific east of Japan (Figure S4d). Previous
studies were very limited to compare with. Meskhidze and Nenes (2006) estimated based on satellite
retrievals a reduction of 15 W m$^{-2}$ in shortwave radiation at TOA due to changes in cloud properties
during a strong phytoplankton bloom event near South Georgia Island in the Southern Ocean in
summertime.

In terms of annual and regional mean, IRE$_{MOA}$ was estimated to be -0.66 W m$^{-2}$ for the western

Pacific, -0.23 W m$^{-2}$ over the EYB region, and -1.04 W m$^{-2}$ over the NWP region, respectively (Table
10). There was an apparent seasonality in the IRE$_{MOA}$, with the maximum of -0.94W m$^{-2}$ in MAM and
the minimum of -0.36 W m$^{-2}$ in JJA over the western Pacific (Table 10). However, the seasonality of
IRE$_{MOA}$ in the EYB and NWP regions are different from that over the western Pacific. Over the EYB,
the estimated IRE$_{MOA}$ reached its maximum (-0.38 W m$^{-2}$) in SON, which was due to the combined
effect of a moderately high MOA concentration (Table 8) and the maximum cloud fraction (Figure S3j)
in this region. Although MOA concentration reached the maximum in MAM, there was a minimum



total cloud fraction in spring among seasons (Figure S3h), leading to a moderate $IRE_{MOA}$. Over the
NWP region, $IRE_{MOA}$ in DJF (-0.57 W m$^{-2}$) was smaller than those in other seasons (-0.78 ~ -1.4 W
m$^{-2}$), which was mainly attributed to the lowest MOA concentration in winter (Table 8), although
cloud fraction was highest among seasons in this region (Figure S3g).

The relative importance of MOA in the aerosol indirect radiative effect over the western Pacific

was investigated by comparing the $IRE_{MOA}$ with the IREs induced by sea salt and anthropogenic
aerosols. In terms of annual and oceanic average, the IREs by sea salt and anthropogenic aerosols
were estimated to be -0.41 W m$^{-2}$ and -7.7 W m$^{-2}$ (Table 10), respectively, indicating $IRE_{MOA}$ (-0.66 W
m$^{-2}$) was larger than the IRE by sea salt and was approximately 9% of that by anthropogenic aerosols.
It is noted that the relative magnitude of $IRE_{MOA}$ compared with the IRE by anthropogenic aerosols
was reduced (5%) in the EYB and enhanced over the NWP (12%), because anthropogenic aerosols
from the East Asian continent dominated aerosol magnitude in the marginal seas of China. In terms of
seasonal and domain average over the western Pacific, $IRE_{MOA}$ was approximately 1.6 times the IRE
by sea salt, and approximately 10% of the IRE by anthropogenic aerosols in MAM. In summer (JJA)
when both sea salt and anthropogenic aerosol concentrations were lowest, $IRE_{MOA}$ was similar in
magnitude to the IRE by sea salt, and about 8% of the IRE by anthropogenic aerosols. In the EYB
region, $IRE_{MOA}$ was just about 5% of the IRE by anthropogenic aerosols, but approximately three
times the IRE by sea salt. In the NWP, $IRE_{MOA}$ was about 12% of the IRE by anthropogenic aerosols
in terms of annual mean, whereas in autumn when anthropogenic aerosol level was relatively low,
$IRE_{MOA}$ was as high as the maximum in MAM and approximately 18% of the IRE by anthropogenic
aerosols. The above model estimation demonstrates that $IRE_{MOA}$ was generally stronger than the IRE
due to sea salt over the western Pacific, and approximately 6~18% of the IRE due to anthropogenic
aerosols in the four seasons over the NWP, suggesting an important role of MOA in perturbing
radiation transfer through modifying cloud properties over the western Pacific Ocean of East Asia.
The estimated IRE due to MSOA accounted for approximately 1% of the annual mean $IRE_{MOA}$
averaged over the western Pacific, consistent with the very low proportion of MSOA in the MOA
mass concentration (Table 8). Overall, MSOA plays a minor role in perturbing cloud properties and
shortwave radiation compared with MPOA.

To address potential uncertainty in the estimated $IRE_{MOA}$ due to limited knowledge on MOA

properties, three additional sensitivity simulations from the base case (results shown in Figure S5 and



Table S4) were carried out regarding particle size, solubility, and molecule weight, which are crucial to aerosol activation (note we focus on MPOA due to its dominant fraction in MOA as shown above). The first sensitivity simulation (SENS1) assumes a smaller geometric mean radius (0.02μm instead of 0.05μm in the base case) for MPOA, resulting in a weaker domain-annual mean $IRE_{MOA}$ (-0.53 W m$^{-2}$) than that in the base case (-0.66 W m$^{-2}$) over the oceanic region (Figure S5b, Table S4). The second sensitivity simulation (SENS2) assigns a lower solubility (0.05) with relatively large molecule weight (146 g mol$^{-1}$) for MPOA (which is similar to the properties of adipic acid, Huff Hartz et al., 2006; Miyazaki et al., 2010) instead of the slight solubility (0.1) with a smaller molecule weight (90 g mol$^{-1}$) (which is similar to the properties of oxalic acid, Roelofs, 2008; Miyazaki et al., 2010) in the base case, in this case, the $IRE_{MOA}$ reduces to -0.2 W m$^{-2}$ (Figure S5c, Table S4). The third simulation (SENS3) combines the above two cases, assuming a smaller geometric mean radius as in SENS1 together with the lower solubility and larger molecule weight as in SENS2, it produces a further reduced $IRE_{MOA}$ of 0.14 W m$^{-2}$ (Figure S5d, Table S4). The above sensitivity simulations exhibit a high sensitivity of $IRE_{MOA}$ to the MPOA properties.

It is interesting to note that Quinn et al. (2017) indicated that sea spray aerosol generally makes a contribution of less than 30% to CCN population at supersaturation of 0.1 to 1.0% on a global basis based on measurements onboard seven research cruises over the Pacific, Southern, Arctic and Atlantic oceans. However, their cruise tracks did not cover the western Pacific Ocean. The results from this study exhibits the annual and oceanic mean contribution of sea spray aerosol (the sum of MOA and sea salt) to the IRE by all aerosols (the sum of MOA, sea salt and anthropogenic aerosols) was approximately 12% in the base case over the western Pacific region (Table 10). This percentage contribution increases to 19% in the NWP in autumn due to the highest $IRE_{MOA}$ and lower IRE by anthropogenic aerosols in autumn (Table 10).

The western Pacific Ocean is just downwind of the East Asian continent, which have large amounts of anthropogenic aerosols, mineral dust, and nutrients inputs to the marginal seas of China from the Yangtze and Yellow Rivers, could be very different from remote clean oceans in the world. Although some cruise measurements have been carried out and a few knowledges on MOA properties (e.g., size distribution) was gained, the cruise measurement and observational analysis for MOA chemical properties are so far almost absent in the western Pacific of East Asia. Therefore, marine biogeochemistry, marine aerosol sources and properties, as well as their potentials to be CCN and



impacts on radiation, cloud, and precipitation deserve further investigation in the future.

4.5 Indirect radiative effect of MOA due to aerosol-radiation and aerosol-cloud interactions
IPCC AR5 proposes the concept of effective radiative forcing (ERF), which is defined as the
change in net radiative flux at TOA after rapid adjustment of the atmosphere (including atmospheric
temperature, water vapor, cloud, circulation) to radiation perturbation with prescribed SST and sea ice,
which can better represent climate response to perturbation of forcing factors (Boucher et al., 2013).
Similarly, this study also estimates the direct radiative effect of MOA due to aerosol-radiation
interaction (denoted as $DRE_{ari}$ hereinafter) and the indirect radiative effect of MOA due to
aerosol-cloud interaction (denoted as $IRE_{aci}$ hereinafter), which consider the adjustment of
atmospheric temperature, water vapor, and cloud (including cloud microphysical and lifetime change,
i.e., the second indirect effect) to the MOA-induced radiation perturbation. $DRE_{ari}$ is derived by the
difference in shortwave radiation flux induced by MOA scattering at TOA between two simulations
with all aerosols and with all aerosols but MOA, while the perturbation of MOA to cloud properties is
turned off. $IRE_{aci}$ is calculated by the difference in shortwave radiation flux resulting from changes in
cloud albedo induced by MOA at TOA between the two simulations, while the perturbation of MOA
to radiation is closed.
Figure 11a shows the annual mean MOA $DRE_{ari}$ in the study domain. It is noted that $DRE_{ari}$ was
not consistently negative in the domain, the effect of atmospheric adjustment on radiation can be seen
in some locations over the Ocean and the continent, with a few positive values of 0.1~0.3 $Wm^{-2}$.
Figure 11b shows the annual mean MOA $IRE_{aci}$ in the study domain, which was similar in
magnitude and distribution pattern to $IRE_{MOA}$ (Figure 10f), but it distributed unevenly in the domain
with some positive values exceeding 0.2 W $m^{-3}$ over both the Western Pacific and the continent. The
atmospheric response and adjustment induced by $IRE_{MOA}$ could be somewhat stronger than that by
$DRE_{MOA}$, given the positive values of $IRE_{aci}$ up to 0.3 W $m^{-3}$ over the continent. The small positive
values could be associated with the radiative feedback and atmospheric and cloud adjustment.
The annual and domain average of $DRE_{ari}$ and $IRE_{aci}$ over the western Pacific are estimated to be
-0.25 $Wm^{-2}$ and -0.61 W $m^{-2}$, respectively, both are somewhat weaker than the $DRE_{MOA}$ (-0.27 $Wm^{-2}$)
and $IRE_{MOA}$ (-0.66 $Wm^{-2}$), which could be due to the offset effect by the positive values.



5. Conclusions.
The organic aerosols of marine origin over the western Pacific Ocean of East Asia was
investigated by an online-coupled regional climate-chemistry model RIEMS-Chem for the year 2014.
Emissions and relevant processes of marine MPOA, isoprene and monoterpene were incorporated into
RIEMS-Chem. A wide variety of observational datasets from EANET, CNEMC and AERONET
networks, cruise measurements and previous publications were collected for model validation. The
modeled SOA from marine VOC sources was also compared with secondary organic tracers measured
by research cruise. The model performed well for $PM_{2.5}$ and $PM_{10}$ in marine environment, producing
overall correlation coefficients and NMBs of 0.61/0.70 and 12%/-7% for $PM_{2.5}$ concentration,
0.65/0.65 and -5%/-1% for $PM_{10}$ concentration at the EANET/CNEMC sites, respectively. The model
reasonably reproduced the spatial distribution and temporal variation of BC and OC concentrations
along cruise tracks and at islands over the west Pacific, with the correlation coefficients and NMBs
being 0.6~0.75 and -28%~3% for OC, respectively. The modeled OC concentration was apparently
improved while taking into account marine organic aerosols. The model results clearly showed an
increasing contribution of marine organic aerosols to total OC mass concentration from the marginal
seas of China to remote oceans. Organic aerosol mass of marine origin was dominated by MPOA
because MSOA produced by marine isoprene and monoterpene emissions was about 1~2 orders of
magnitude lower than MPOA. The model simulates AOD reasonably well at the 7 coastal or island
AERONET sites, with an overall correlation coefficient of 0.54 and an NMB of -6%.
High MPOA emission mainly occurred over the marginal seas of China (EYB) and the northern
parts of western Pacific northeast of Japan (NWP). For the western Pacific, MPOA emission reached
the maximum in SON, followed by those in DJF and MAM, and the minimum in JJA, with an annual
and domain average emission rate of $0.16 \times 10^{-2}$ µg m$^{-2}$ s$^{-1}$. The combination of Chl-a concentration and
sea salt emission flux determined the seasonality of MPOA emission. The annual MPOA emission for
the year 2014 was estimated to be 0.78 Tg yr$^{-1}$ over the western Pacific.
Consistent with the distribution pattern MPOA emission, high MOA concentration mainly
distributed over the EYB and NWP, with an annual and domain mean concentration of 0.27 µg m$^{-3}$,
0.48 µg m$^{-3}$ and 0.59 µg m$^{-3}$, over the western Pacific, the EYB and NWP regions, respectively. MOA
concentration was highest in MAM and lowest in DJF, with the seasonal and domain mean values of
0.37 µg m$^{-3}$ and 0.21 µg m$^{-3}$, respectively, over the western Pacific. The seasonality of MOA



concentration was determined by the combined effect of MPOA emission, dry and wet depositions.

On average, the annual mean percentage contribution of MOA to total OA mass was 26% over the western Pacific, with the largest seasonal mean contribution of 32% in SON and the lower ones in DJF (24%) and JJA (23%). Over the NWP, the domain average contribution of MOA to OA could be as high as 42% in terms of annual mean and approaching 52% in MAM; however, over the EYB, the annual mean contribution was just 13% and the percentage contribution was even reduced to 6% in JJA. This indicated that the relative importance of MOA in total OA concentration increased with the distance away from the East Asian continent. MSOA concentration was approximately 1~2 orders of magnitude lower than MPOA, with the simulated annual and regional mean MSOA being 2.2 ng m$^{-3}$ and the maximum daily mean value up to 28 ng m$^{-3}$ in summer over the western Pacific.

An annual/oceanic mean all-sky DRE$_{MOA}$ of -0.27 W m$^{-2}$ at TOA was estimated over the western Pacific, which was about 40% of the IRE$_{MOA}$ (-0.66 W m$^{-2}$). The domain mean IRE$_{MOA}$ was strongest in spring (-0.94 W m$^{-2}$) and weakest in summer (-0.36 W m$^{-2}$) over the western Pacific, and the monthly mean IRE$_{MOA}$ can reach -5 W m$^{-2}$ in the NWP region east of Japan in April. The changes in MOA concentration and cloud amount both contributed to the seasonality of IRE$_{MOA}$. In terms of annual and oceanic mean over the western Pacific, MSOA just contributed approximately 1% of the IRE$_{MOA}$. IRE$_{MOA}$ was generally larger than the IRE due to sea salt on average. The annual and oceanic mean IRE due to sea spray aerosols (MOA + sea salt) was approximately 12% of that due to all aerosols (anthropogenic + MOA + sea salt aerosols) over the western Pacific, but this ratio can increase up to 19% in autumn in the NWP region. The estimation of IRE$_{MOA}$ was sensitive to MOA properties, which decreased apparently while a smaller geometric mean radius together with a lower solubility and a larger molecule weight were assigned for MOA. Overall, the indirect radiative effect of MOA was larger than the direct radiative effect, and had a nonnegligible impact on radiation budget and cloud over the western Pacific. The direct and indirect radiative effect considering atmospheric feedback and adjustment were estimated as well, which was similar in magnitude to the DRE$_{MOA}$ and IRE$_{MOA}$, with a few positive changes in shortwave radiation fluxes in some locations.

While this study presents new insights into the seasonal variation and annual means of emissions, concentrations, and radiative effects of MOA in the western Pacific, it is still subject to some uncertainties as follows: 1.) the properties of marine organic aerosols, including size distribution, molecular weight, solubility, surfactant amount etc. are still poorly characterized, which are crucial to



aerosol activation, dry deposition, and wet scavenging; 2.) the sources and chemical formation

processes of marine organic aerosols including secondary organics are highly complex, and poorly

understood and represented in the model; 3.) the indirect effects of MOA in this study is for warm

stratiform cloud. Further research on MOA sources, properties, chemical processes, and climatic

impacts will be conducted together with the advances in both field experiments (integrated cruise,

aircraft and satellite observations) and model development in the future.

**Author Contributions.**

ZH designed the study, JL and ZH developed the model, processed and analyzed the model results, JL

performed the model simulation, ZH and JL wrote the paper, PF and XY provided and analyzed the

cruise measurement data.

**Data availability.**

The observational data can be accessed through contacting the corresponding author.

**Competing interests.**

The authors declare that they have no conflict of interests.

**Acknowledgement.**

This study was supported by the National Key R&D Program of China (No. 2019YFA0606802),

the National Natural Science Foundation of China (No. 42275118, No. 91644217), the Jiangsu

Collaborative Innovation Center for Climate Change. The authors appreciate the science teams of

EANET, CNEMC, AERONET, and VIIRS for their works in data maintenance.

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





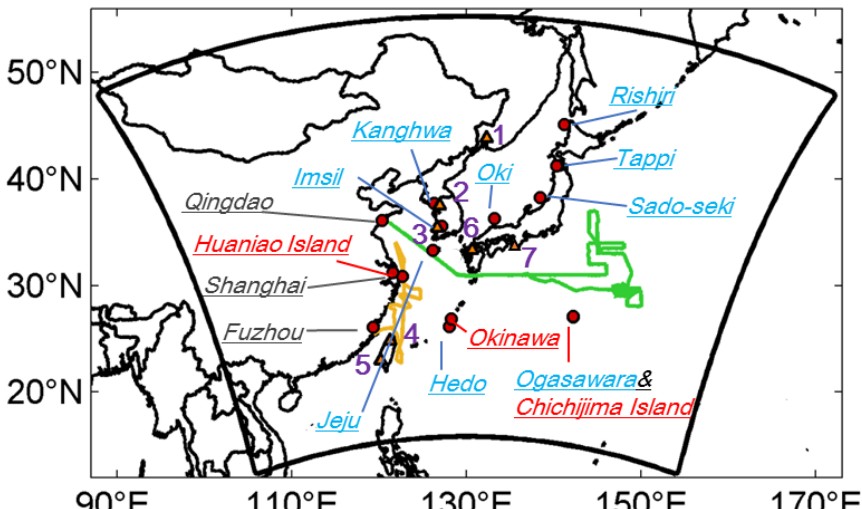

Figure 1. Model domain, observational sites, and research cruise tracks. EANET sites are marked in light-blue. Observation sites of carbonaceous aerosols are marked in red (Chichijima Island: Boreddy et al., 2018; Fukue: Kanaya et al., 2016; Okinawa: Kunwar and Kawamura, 2014; Huaniao Island: Wang F. W. et al., 2014). Three CNEMC sites are marked in grey (Qingdao, Shanghai, and Fuzhou). Two research cruise tracks are represented by green line (Dongfanghong II from 17 March to 22 April 2014: Luo et al., 2016; Feng et al., 2017) and orange line (KEXUE-1 from 18 May to 12 June 2014: Kang et al., 2018), respectively. AERONET sites are represented by triangles with numbers (1-Ussuriysk, 2-Yonsei_University, 3-Gwangju_GIST, 4-EPA-NCU, 5-Chen-Kung_Univ, 6-Fukuoka, and 7-Shirahama). Full names of abbreviations are given in the text.



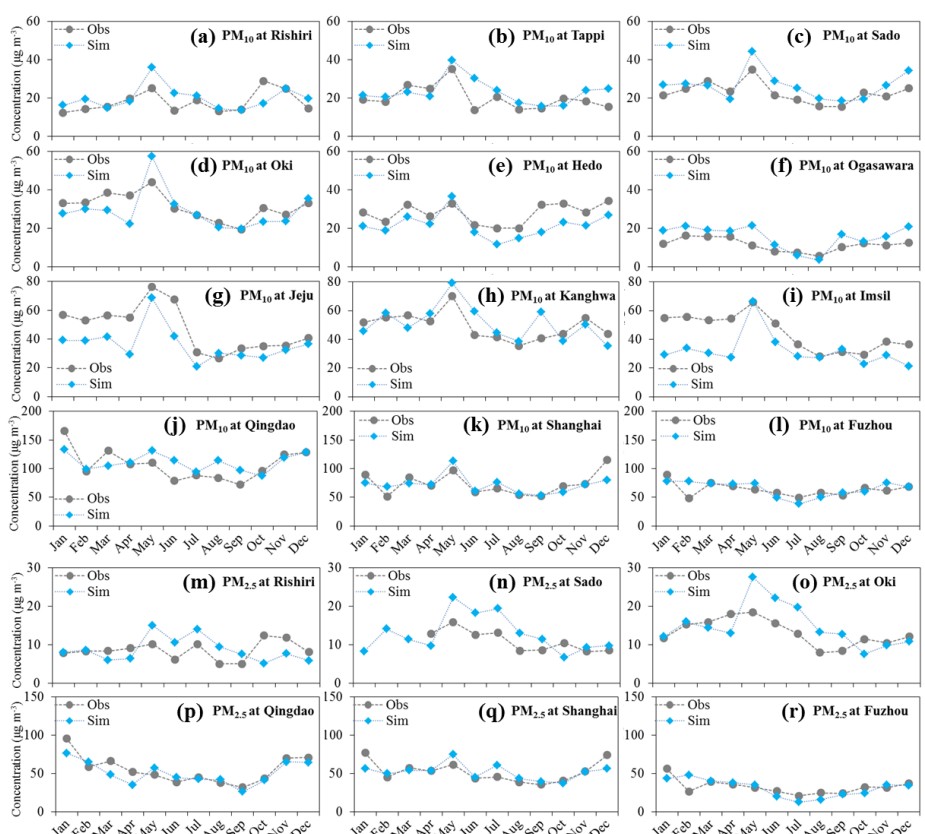

Figure 2. The model simulated (Sim) and observed (Obs) monthly PM$_{10}$ (a~l) and PM$_{2.5}$ (m~r) concentrations at EANET and CNEMC sites for the year 2014. The monthly data were averaged from hourly observations and the simulations were sampled according to the observations.

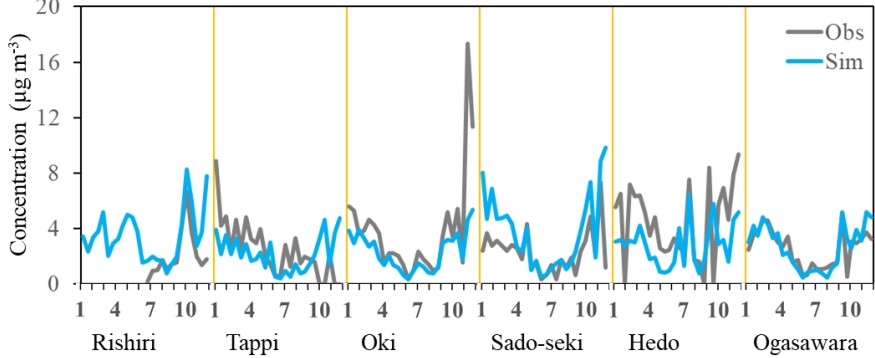

Figure 3. Observed and model simulated sodium (Na+) concentrations (bi-weekly samples) at 6 coastal/island EANET sites in Japan for the year 2014. The x axis is month for each site.





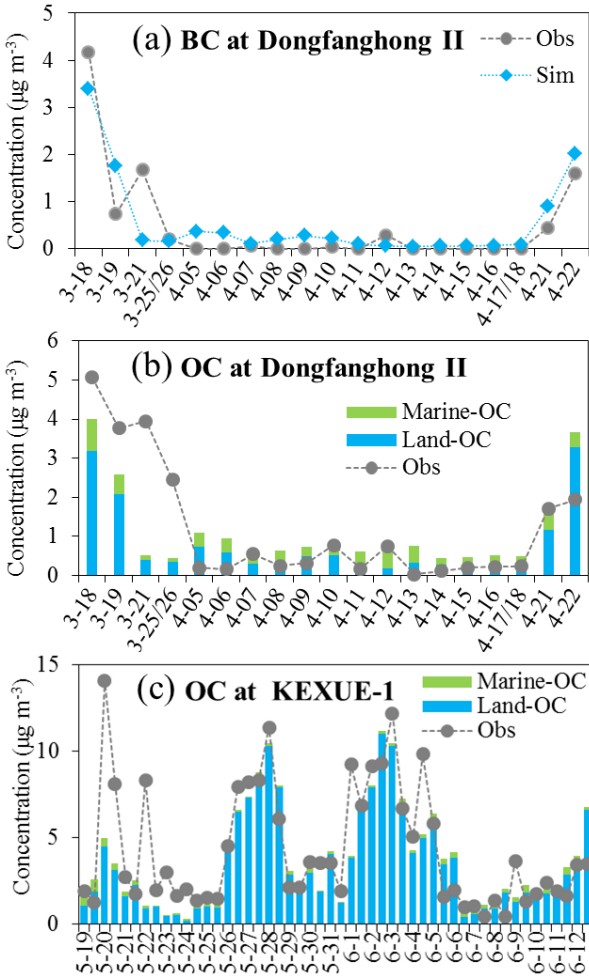

Figure 4. The model simulated (bars) and observed (dotted lines) daily BC and OC concentrations from the spring campaign (a, b) and half-day OC concentrations from the early summer campaign (c). The modeled total OC concentration was decomposed into those from marine (green bars) and land (blue bars) sources.



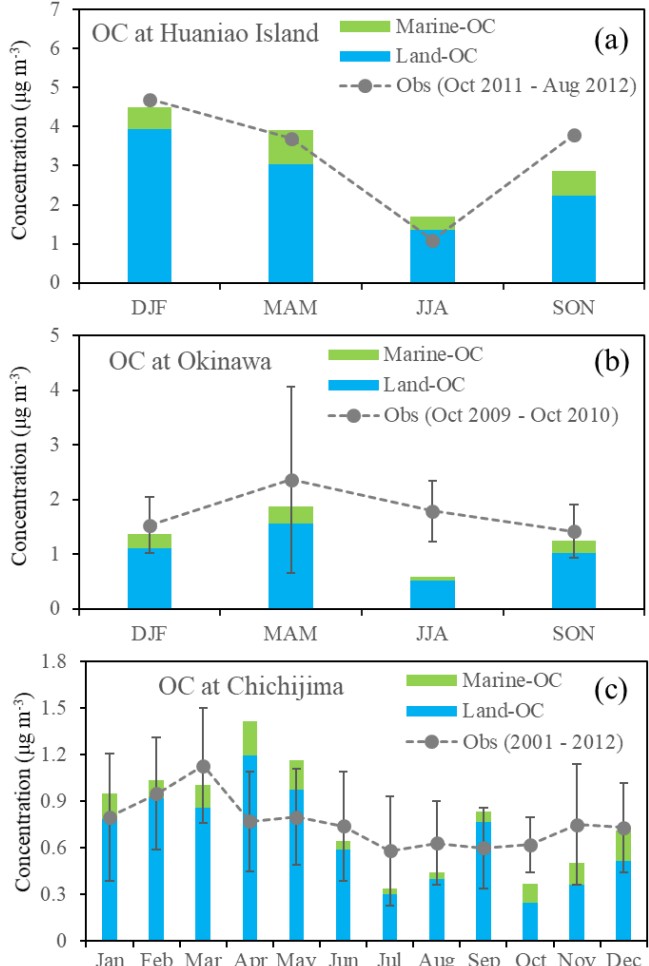

Figure 5. The model simulated (bars) and observed (dotted lines) OC concentrations at different sites. Seasonal mean concentrations were provided at (a) Huaniao Island (Wang et al., 2015) and (b) Okinawa (Kunwar and Kawamura, 2014) while monthly mean concentrations were provided at (c) Chichijima Island (Boreddy et al., 2018). Standard deviations were available at Okinawa and Chichijima. The modeled OC concentrations were decomposed to marine (green bars) and land (blue bars) sources. The simulation is for the year 2014.





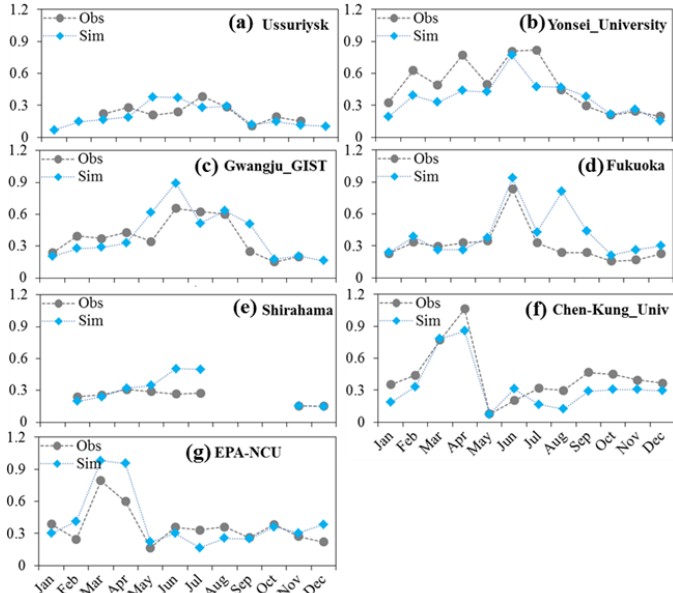

Figure 6. The model simulated (Sim) and observed (Obs) monthly mean AOD at 7 AERONET sites for the year 2014. The monthly mean observations were calculated from hourly data and the corresponding simulations were sampled according to the observations.





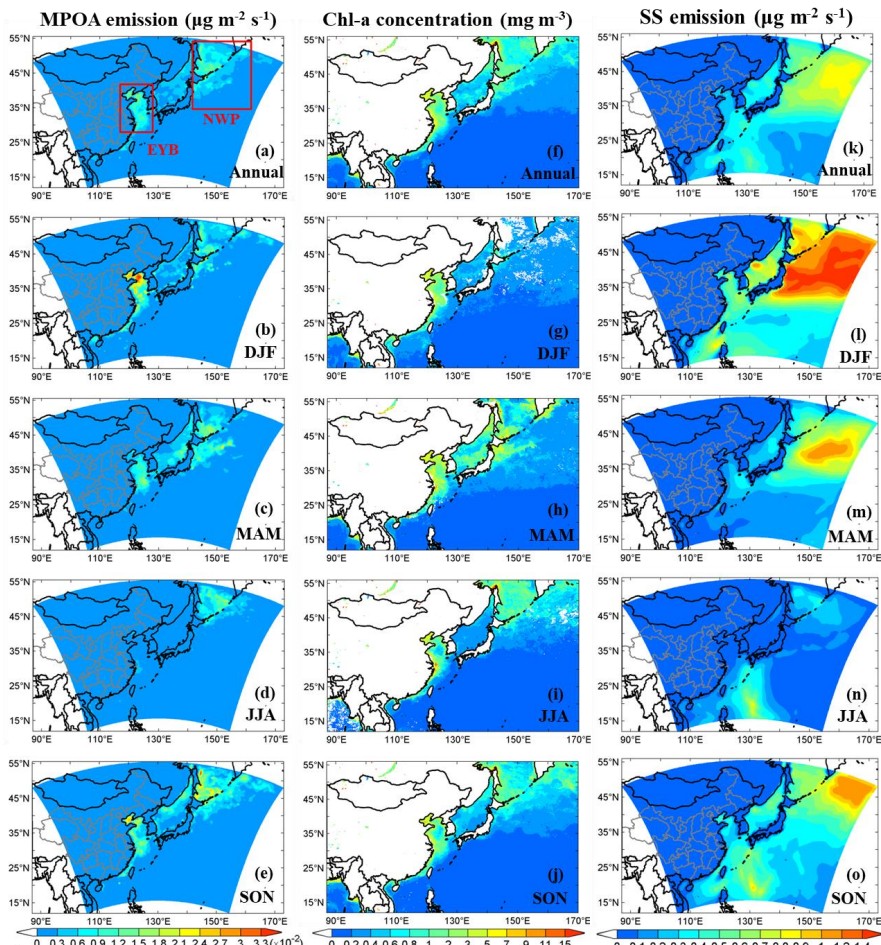

Figure 7. Model simulated annual and seasonal mean distributions of MPOA emissions (a~e), VIIRS retrieved surface sea water chlorophyll-a (Chl-a) concentrations (f~j), and model simulated sea salt (SS) emissions (k~o). Two hotspot regions are marked with red boxes: the region including the East China Sea, the Yellow Sea, and the Bohai Sea (EYB, 27~40°N, 115~123°E) and the region including the northern parts of the western Pacific to the northeast of Japan (NWP, 35~55°N, 140~160°E). Units are given in parentheses.



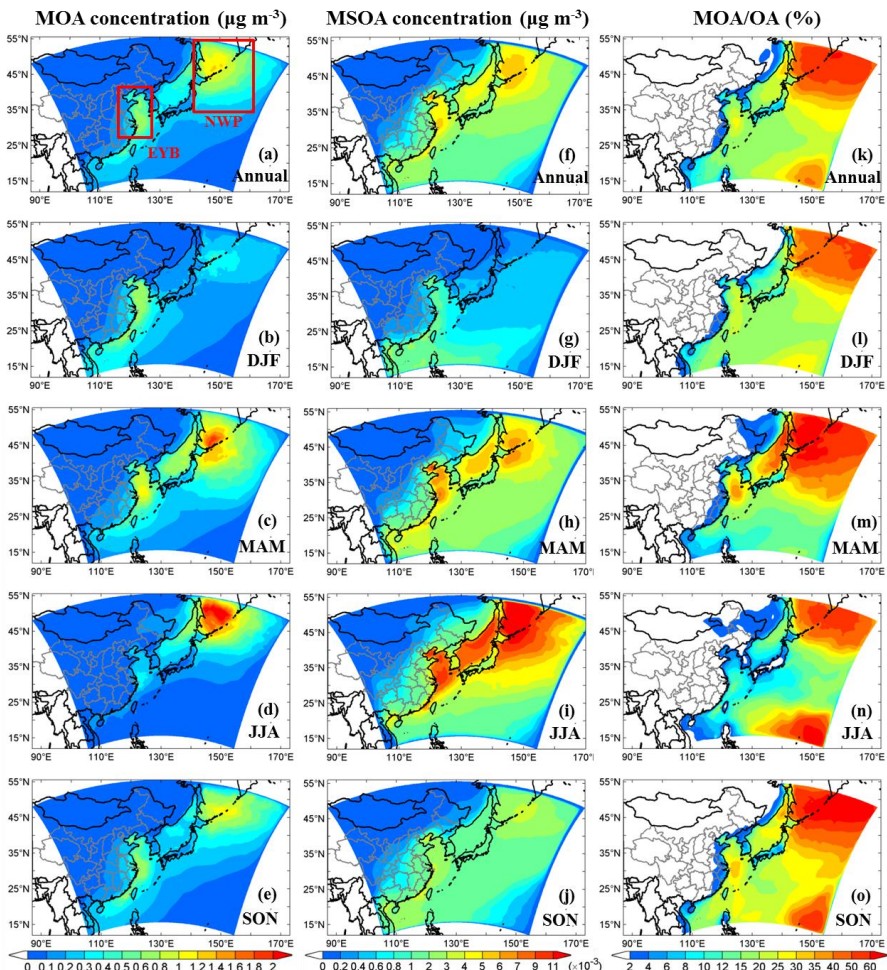

Figure 8. Model simulated annual and seasonal mean near surface MOA (primary+secondary) concentrations (a~e), near surface MSOA concentrations (f~j), and percentage contributions of MOA to total OA (k~o). The two regions of the EYB (27~40°N, 115~123°E) and the NWP (35~55°N, 140~160°E) are marked in 8a. Units are given in parentheses.



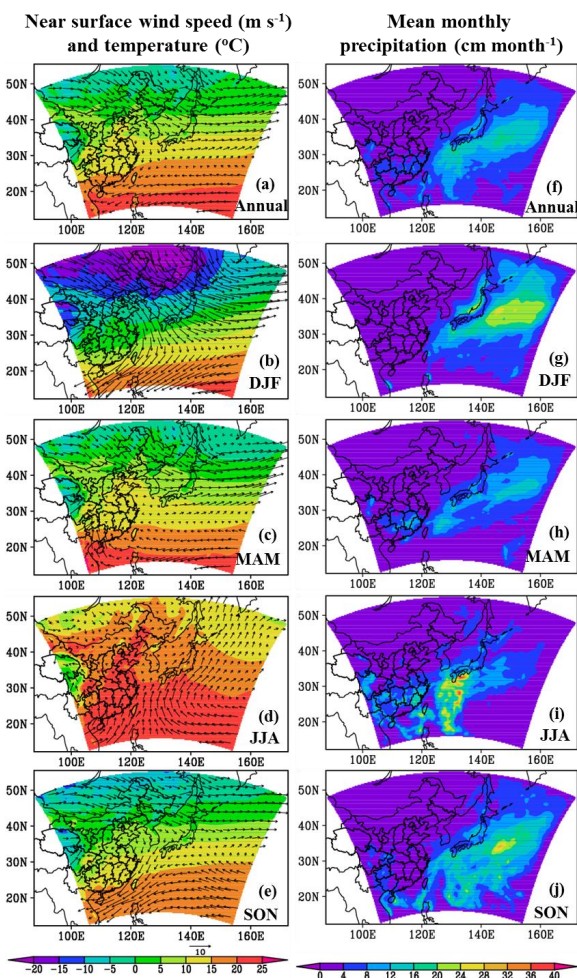

Figure 9. Model simulated annual and seasonal mean near surface temperatures (unite: °C) overlaid with wind vectors (unit: m s⁻¹) (a~e) and mean monthly precipitations (unit: cm month⁻¹) (f~j).



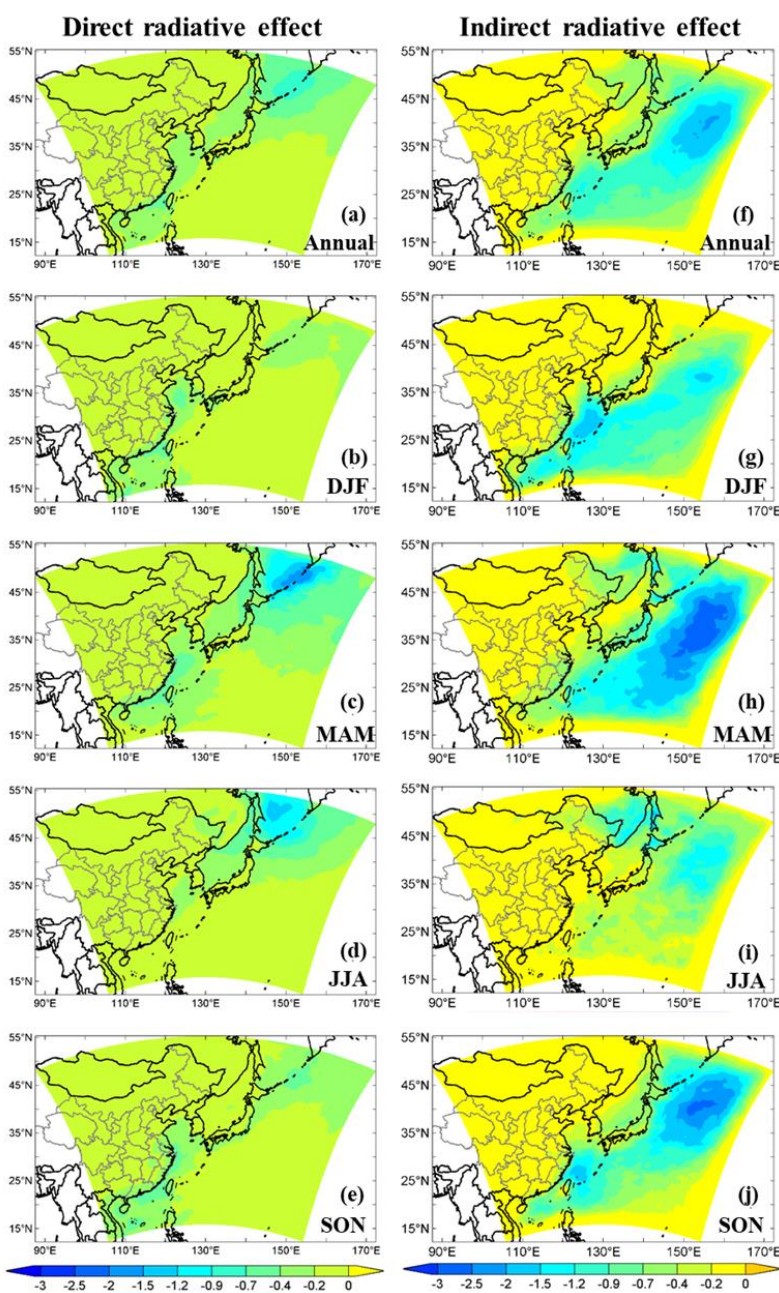

Figure 10. Model simulated annual and seasonal mean direct radiative effect due to MOA (DRE$_{MOA}$) (a~e) and indirect radiative effect due to MOA (IRE$_{MOA}$) (f~j) at the top of atmosphere (TOA) under all-sky condition (unit: W m$^{-2}$).





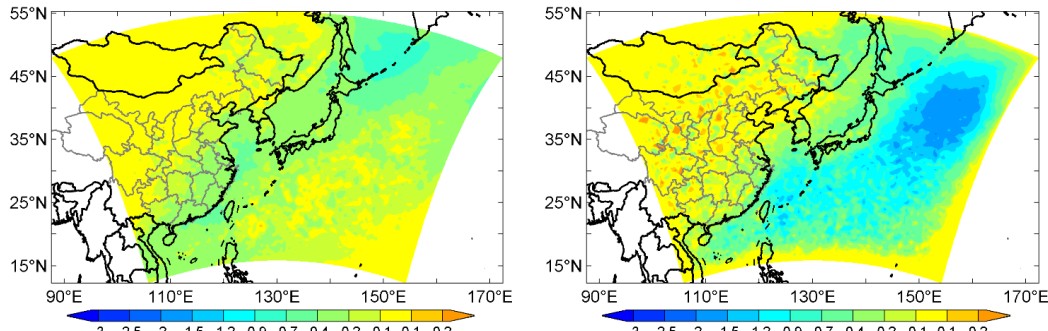

Figure 11. Model simulated annual mean (a) direct radiative effect of MOA due to aerosol-radiation interaction, (b) indirect radiative effect of MOA due to aerosol-cloud interaction at the top of atmosphere (TOA) (unit: W m$^{-2}$)





Table 1. Annual and seasonal performance statistics for hourly $PM_{10}$ and $PM_{2.5}$ concentrations (unit: µg m$^{-3}$) at EANET sites for the year 2014. Mean observation (Obs), mean simulation (Sim), correlation coefficient (R), and normalized mean bias (NMB in %) are listed. ANN=annual, DJF=December-January-February, MAM=March-April-May, JJA=June-July-August, and SON=September-October-November.

| Sites | Samples | ANN | | | | DJF | | | | MAM | | | | JJA | | | | SON | | | |
|---|---|---|---|---|---|---|---|---|---|---|---|---|---|---|---|---|---|---|---|---|---|
| | | Obs | Sim | R | NMB | Obs | Sim | R | NMB | Obs | Sim | R | NMB | Obs | Sim | R | NMB | Obs | Sim | R | NMB |
| **PM$_{10}$** | | | | | | | | | | | | | | | | | | | | | |
| Rishiri | 8381 | 18.0 | 19.9 | 0.53 | 11 | 13.6 | 18.4 | 0.65 | 35 | 20.1 | 23.0 | 0.56 | 15 | 15.2 | 19.4 | 0.42 | 28 | 23.0 | 18.8 | 0.51 | -18 |
| Tappi | 8584 | 20.1 | 23.2 | 0.49 | 15 | 17.4 | 22.3 | 0.54 | 28 | 29.0 | 28.0 | 0.59 | -4 | 16.1 | 23.8 | 0.18 | 48 | 17.6 | 18.5 | 0.39 | 5 |
| Sado | 8640 | 22.8 | 26.4 | 0.63 | 16 | 23.6 | 29.5 | 0.68 | 25 | 29.2 | 30.3 | 0.65 | 4 | 18.6 | 24.5 | 0.55 | 32 | 19.6 | 21.4 | 0.53 | 9 |
| Oki | 8424 | 31.3 | 29.2 | 0.68 | -7 | 33.2 | 31.1 | 0.65 | -7 | 40.2 | 37.5 | 0.71 | -7 | 26.7 | 26.6 | 0.61 | 0 | 25.8 | 22.3 | 0.66 | -14 |
| Hedo | 8008 | 27.7 | 21.7 | 0.56 | -22 | 28.8 | 22.4 | 0.66 | -22 | 30.5 | 28.3 | 0.58 | -7 | 20.7 | 14.8 | 0.54 | -28 | 30.9 | 20.8 | 0.34 | -33 |
| Ogasawara | 8120 | 11.5 | 15.7 | 0.48 | 36 | 13.4 | 20.3 | 0.38 | 52 | 14.2 | 19.7 | 0.40 | 39 | 7.0 | 6.8 | 0.46 | -2 | 11.2 | 15.0 | 0.30 | 34 |
| Jeju | 7101 | 46.9 | 36.9 | 0.64 | -21 | 50.1 | 38.2 | 0.71 | -24 | 62.6 | 46.6 | 0.66 | -26 | 36.4 | 31.5 | 0.44 | -13 | 34.7 | 29.3 | 0.44 | -15 |
| Kanghwa | 8524 | 49.2 | 51.2 | 0.59 | 4 | 50.2 | 46.0 | 0.60 | -8 | 59.9 | 61.9 | 0.66 | 3 | 40.0 | 47.2 | 0.47 | 18 | 46.5 | 49.3 | 0.38 | 6 |
| Imsil | 8383 | 44.5 | 32.3 | 0.58 | -27 | 48.8 | 27.9 | 0.63 | -43 | 58.0 | 42.1 | 0.62 | -27 | 38.4 | 31.1 | 0.47 | -19 | 33.0 | 28.2 | 0.42 | -15 |
| Average | 74165 | 30.0 | 28.5 | 0.65 | -5 | 30.8 | 28.3 | 0.67 | -8 | 37.9 | 35.2 | 0.65 | -7 | 23.9 | 25.1 | 0.59 | 5 | 26.9 | 25.0 | 0.58 | -7 |
| **PM$_{2.5}$** | | | | | | | | | | | | | | | | | | | | | |
| Rishiri | 8331 | 8.6 | 8.7 | 0.54 | 0 | | | | | | | | | | | | | | | | |
| Sado | 6517 | 11.0 | 13.4 | 0.53 | 21 | 8.1 | 7.4 | 0.78 | -8 | 9.2 | 9.2 | 0.56 | 0 | 7.2 | 11.5 | 0.54 | 59 | 10.0 | 6.7 | 0.31 | -33 |
| Oki | 8410 | 13.1 | 15.0 | 0.64 | 14 | 8.5 | 9.8 | 0.60 | 14 | 14.4 | 16.1 | 0.63 | 12 | 11.4 | 16.8 | 0.47 | 48 | 9.1 | 9.1 | 0.24 | 0 |
| Average | 23258 | 10.9 | 12.3 | 0.61 | 12 | 13.0 | 12.9 | 0.77 | -1 | 17.4 | 18.7 | 0.64 | 8 | 12.1 | 18.3 | 0.55 | 51 | 10.1 | 10.0 | 0.39 | -1 |





Table 2. Same as Table 1 but for CNEMC sites.

| Sites | Samples | ANN | | | | DJF | | | | MAM | | | | JJA | | | | SON | | | |
|---|---|---|---|---|---|---|---|---|---|---|---|---|---|---|---|---|---|---|---|---|---|
| | | Obs | Sim | R | NMB | Obs | Sim | R | NMB | Obs | Sim | R | NMB | Obs | Sim | R | NMB | Obs | Sim | R | NMB |
| **PM$_{10}$** | | | | | | | | | | | | | | | | | | | | | |
| Qingdao | 7622 | 107.0 | 108.6 | 0.61 | 1 | 131.0 | 124.3 | 0.76 | -5 | 117.3 | 109.9 | 0.49 | -6 | 83.6 | 108.4 | 0.64 | 30 | 97.1 | 101.4 | 0.59 | 4 |
| Shanghai | 7581 | 73.4 | 70.5 | 0.55 | -4 | 93.9 | 81.1 | 0.72 | -14 | 83.2 | 80.2 | 0.58 | -4 | 59.6 | 64.0 | 0.37 | 7 | 64.8 | 60.6 | 0.43 | -7 |
| Fuzhou | 7610 | 63.7 | 63.8 | 0.38 | 0 | 69.9 | 72.9 | 0.30 | 4 | 69.8 | 72.6 | 0.32 | 4 | 55.3 | 45.8 | 0.28 | -17 | 60.5 | 64.5 | 0.30 | 7 |
| Average | 22813 | 81.6 | 80.7 | 0.65 | -1 | 98.4 | 92.6 | 0.74 | -6 | 89.9 | 86.7 | 0.58 | -4 | 66.0 | 72.0 | 0.61 | 9 | 74.1 | 75.4 | 0.51 | 2 |
| **PM$_{2.5}$** | | | | | | | | | | | | | | | | | | | | | |
| Qingdao | 7627 | 55.2 | 48.7 | 0.72 | -12 | 75.1 | 67.8 | 0.83 | -10 | 56.3 | 43.7 | 0.61 | -22 | 40.5 | 42.8 | 0.60 | 6 | 48.2 | 43.9 | 0.74 | -9 |
| Shanghai | 7724 | 51.9 | 51.8 | 0.62 | 0 | 68.0 | 59.6 | 0.80 | -12 | 57.2 | 57.5 | 0.60 | 0 | 42.6 | 49.8 | 0.46 | 17 | 42.6 | 42.9 | 0.51 | 1 |
| Fuzhou | 7641 | 32.3 | 30.0 | 0.44 | -7 | 40.3 | 40.2 | 0.25 | 0 | 35.8 | 36.8 | 0.37 | 3 | 24.0 | 15.8 | 0.38 | -34 | 29.2 | 27.3 | 0.29 | -7 |
| Average | 22992 | 46.6 | 43.4 | 0.70 | -7 | 61.1 | 55.8 | 0.78 | -9 | 49.7 | 45.6 | 0.63 | -8 | 35.6 | 35.5 | 0.62 | 0 | 39.9 | 38.0 | 0.62 | -5 |



Table 3. Performance statistics for BC and OC from the two research campaigns in 2014. BC and OC were measured on Dongfanghong II during the spring campaign whereas only OC were collected on KEXUE-1 during the early summer campaign. Mean observation (Obs), mean simulation (Sim), correlation coefficient (R), and normalized mean bias (NMB in %) are listed. The modeled concentrations of marine-OC (including MPOA and MSOA) and its contribution to total OC were estimated.

| | Dongfanghong II | | | KEXUE-1 | |
|---|---|---|---|---|---|
| | BC | OC | Marine-OC (% in OC) | OC | Marine-OC (% in OC) |
| Samples | 19 | 19 | | 51 | |
| Obs ($\mu g\ m^{-3}$) | 0.49 | 1.20 | | 4.26 | |
| Sim ($\mu g\ m^{-3}$) | 0.55 | 1.14 | 0.33 (29%) | 3.68 | 0.23 (6%) |
| R | 0.87 | 0.66 | | 0.75 | |
| NMB (%) | 13 | -5 | | -13 | |



Table 4. Comparison of model simulated and observed seasonal OC concentrations (unit: μg m$^{-3}$) at Huaniao Island and Okinawa. The modeled concentrations of marine-OC and its contribution to total OC were estimated. ANN=annual, DJF=December-January-February, MAM=March-April-May, JJA=June-July-August, and SON=September-October-November.

| | | Time | ANN[c] | DJF | MAM | JJA | SON | Reference |
|---|---|---|---|---|---|---|---|---|
| **OC** | | | | | | | | |
| Huaniao Island | Obs | Oct 2011~ Aug 2012 | 3.3 | 4.7 | 3.7 | 1.1 | 3.8 | Wang F. W. et al., 2015 |
| | Sim | 2014 | 3.2 | 4.5 | 3.9 | 1.7 | 2.9 | |
| | Marine-OC | | 0.6 | 0.56 | 0.88 | 0.32 | 0.65 | |
| | (% in OC) | | (19%) | (12%) | (22%) | (19%) | (23%) | |
| Okinawa | Obs | Oct 2009~ Oct 2010 | 1.8 | 1.5 | 2.4 | 1.8 | 1.4 | Kunwar and Kawamura, 2014 |
| | Sim | 2014 | 1.3 | 1.4 | 1.9 | 0.6 | 1.2 | |
| | Marine-OC | | 0.21 | 0.25 | 0.32 | 0.06 | 0.23 | |
| | (% in OC) | | (17%) | (18%) | (17%) | (10%) | (18%) | |

a: The location of Huaniao Island is 30.86°N, 122.67°E.

b: The location of Okinawa Island is 26.15°N, 128.03°E.

c: The annual means are averages of the four seasonal means.





Table 5. Comparison of model simulated and observed monthly mean OC concentrations (unit: μg m$^{-3}$) at Chichijima Island. Marine-OC concentration and its contribution to total OC were estimated.

| Month | Jan | Feb | Mar | Apr | May | Jun | Jul | Aug | Sep | Oct | Nov | Dec | Annual |
|---|---|---|---|---|---|---|---|---|---|---|---|---|---|
| Obs[a] | 0.80 | 0.95 | 1.13 | 0.77 | 0.80 | 0.74 | 0.58 | 0.63 | 0.60 | 0.62 | 0.75 | 0.73 | 0.76 |
| Sim[b] | 0.95 | 1.03 | 1.01 | 1.42 | 1.17 | 0.64 | 0.34 | 0.44 | 0.84 | 0.37 | 0.50 | 0.71 | 0.78 |
| Marine-OC | 0.17 | 0.11 | 0.15 | 0.22 | 0.19 | 0.06 | 0.04 | 0.04 | 0.07 | 0.12 | 0.14 | 0.19 | 0.13 |
| (% in OC) | (18%) | (11%) | (15%) | (16%) | (16%) | (9%) | (11%) | (9%) | (8%) | (33%) | (28%) | (27%) | (16%) |

a: Observations at Chichijima Island (27.07°N, 142.22°E) were obtained from Boreddy et al. (2018) and are 12-yr averages (2001-2012).

b: Simulations are for the year 2014.



Table 6. Performance statistics for hourly AOD (unitless) at AERONET sites for the year 2014. Mean observation (Obs), mean simulation (Sim), correlation coefficient (R), and normalized mean bias (NMB in %) are listed. IDs are marked in Figure 1.

| ID | Site | Obs | Sim | R | NMB | Samples |
|---|---|---|---|---|---|---|
| 1 | Ussuriysk | 0.22 | 0.21 | 0.41 | -6 | 945 |
| 2 | Yonsei_University | 0.48 | 0.37 | 0.67 | -23 | 1629 |
| 3 | Gwangju_GIST | 0.33 | 0.36 | 0.53 | 7 | 900 |
| 4 | EPA-NCU | 0.38 | 0.39 | 0.43 | 4 | 685 |
| 5 | Chen-Kung_Univ | 0.49 | 0.37 | 0.60 | -25 | 657 |
| 6 | Fukuoka | 0.28 | 0.34 | 0.50 | 18 | 1144 |
| 7 | Shirahama | 0.26 | 0.31 | 0.40 | 19 | 752 |
|  | Average | 0.36 | 0.34 | 0.54 | -6 | 6712 |

Table 7. Modeled domain and annual/seasonal mean MPOA emission rates, surface sea water chlorophyll-a (Chl-a) concentrations, and sea salt emission fluxes over the western Pacific of East Asia (Mean), the region including the East China Sea, the Yellow Sea, and the Bohai Sea (EYB) and the region including northern parts of western Pacific to the northeast of Japan (NWP).

| | MPOA emission ($\times10^{-2}$ µg m$^{-2}$ s$^{-1}$) | | | | Chl-a concentration (mg m$^{-3}$) | | | Sea salt emission flux (µg m$^{-2}$ s$^{-1}$) | | |
|---|---|---|---|---|---|---|---|---|---|---|
| | Mean[a] | Max[b] | EYB[c] | NWP[d] | Mean[a] | EYB[c] | NWP[d] | Mean[a] | EYB[c] | NWP[d] |
| ANN | 0.16 | 1.8 | 0.65 | 0.40 | 1.17 | 3.51 | 0.96 | 0.36 | 0.18 | 0.59 |
| DJF | 0.18 | 3.6 | 1.19 | 0.33 | 0.67 | 3.20 | 0.37 | 0.63 | 0.35 | 1.09 |
| MAM | 0.17 | 2.5 | 0.41 | 0.43 | 0.97 | 4.00 | 1.13 | 0.30 | 0.11 | 0.61 |
| JJA | 0.08 | 1.9 | 0.12 | 0.29 | 1.07 | 3.14 | 0.90 | 0.14 | 0.04 | 0.15 |
| SON | 0.20 | 3.5 | 0.88 | 0.54 | 1.10 | 2.90 | 0.90 | 0.38 | 0.24 | 0.53 |

a: Mean over oceanic areas.

b: Maximums over oceanic areas.

c: Ocean areas within 27~40°N, 115~123°E.

d: Ocean areas within 35~55°N, 140~160°E.



Table 8. Modeled domain and annual/seasonal mean near surface MOA concentrations, MSOA concentrations, and MOA to total OA ratios over the western Pacific of East Asia (Mean), the EYB region, and the NWP region.

| | MOA concentration (µg m$^{-3}$) | | | | MSOA concentration (×10$^{-3}$ µg m$^{-3}$) | | | | MOA/OA (%) | | | |
|---|---|---|---|---|---|---|---|---|---|---|---|---|
| | Mean[a] | Max[b] | EYB[c] | NWP[d] | Mean[a] | Max[b] | EYB[c] | NWP[d] | Mean[a] | Max[b] | EYB[c] | NWP[d] |
| ANN | 0.27 | 1.2 | 0.48 | 0.59 | 2.2 | 6.9 | 4.1 | 3.8 | 26% | 62% | 13% | 42% |
| DJF | 0.21 | 0.8 | 0.54 | 0.23 | 0.7 | 3.2 | 1.0 | 0.4 | 24% | 57% | 11% | 36% |
| MAM | 0.37 | 1.9 | 0.62 | 0.81 | 2.7 | 10.5 | 5.3 | 4.1 | 26% | 69% | 15% | 52% |
| JJA | 0.23 | 2.3 | 0.22 | 0.8 | 3.9 | 13.6 | 7.5 | 8.3 | 23% | 69% | 6% | 32% |
| SON | 0.26 | 1.3 | 0.52 | 0.52 | 1.5 | 4.2 | 2.6 | 2.2 | 32% | 73% | 18% | 48% |

a: Mean over oceanic areas.

b: Maximums over oceanic areas.

c: Ocean areas within 27~40°N, 115~123°E.

d: Ocean areas within 35~55°N, 140~160°E.

Table 9. Modeled domain and annual/seasonal mean near surface wind speed, temperature, precipitation, and relative humidity (RH) over the western Pacific of East Asia (Mean), the EYB region, and the NWP region.

| | Wind speed (m s$^{-1}$) | | | Temperature (°C) | | | Precipitation (cm month$^{-1}$) | | | RH (%) | | |
|---|---|---|---|---|---|---|---|---|---|---|---|---|
| | Mean[a] | EYB[b] | NWP[c] | Mean[a] | EYB[b] | NWP[c] | Mean[a] | EYB[b] | NWP[c] | Mean[a] | EYB[b] | NWP[c] |
| ANN | 4.3 | 2.9 | 4.0 | 19.2 | 15.1 | 8.5 | 6.1 | 2.7 | 8.0 | 78 | 73 | 83 |
| DJF | 6.4 | 4.5 | 6.9 | 14.0 | 4.5 | 1.0 | 7.0 | 1.8 | 12.4 | 75 | 67 | 77 |
| MAM | 3.8 | 2.0 | 3.7 | 16.9 | 13.4 | 5.1 | 4.3 | 2.1 | 7.0 | 79 | 75 | 84 |
| JJA | 3.0 | 1.9 | 2.5 | 24.0 | 23.2 | 15.8 | 5.1 | 3.5 | 3.7 | 83 | 80 | 94 |
| SON | 4.1 | 3.1 | 3.1 | 21.7 | 17.9 | 12.0 | 7.9 | 3.2 | 9.0 | 76 | 71 | 77 |

a: Mean over oceanic areas.

b: Ocean areas within 27~40°N, 115~123°E.

c: Ocean areas within 35~55°N, 140~160°E.



Table 10. Modeled regional and annual/seasonal mean all-sky TOA direct radiative effect (DRE) and indirect radiative effects (IRE) due to MOA, anthropogenic aerosols, and sea salt over oceanic areas of the western Pacific (WP), the EYB region, and the NWP region. The units are W m$^{-2}$.

| | MOA | | | Anthropogenic | | | Sea salt | | |
|---|---|---|---|---|---|---|---|---|---|
| | WP[a] | EYB[b] | NWP[c] | WP[a] | EYB[b] | NWP[c] | WP[a] | EYB[b] | NWP[c] |
| | | | | | DRE | | | | |
| ANN | -0.27 | -0.33 | -0.50 | -2.8 | -6.6 | -2.7 | -0.86 | -0.56 | -0.89 |
| DJF | -0.18 | -0.32 | -0.21 | -1.2 | -3.5 | -1.2 | -0.93 | -0.48 | -0.95 |
| MAM | -0.38 | -0.34 | -0.76 | -2.8 | -6.2 | -3.0 | -0.79 | -0.36 | -1.09 |
| JJA | -0.28 | -0.26 | -0.72 | -4.9 | -11.0 | -5.0 | -0.77 | -0.85 | -0.79 |
| SON | -0.26 | -0.38 | -0.32 | -2.2 | -5.5 | -1.4 | -0.94 | -0.55 | -0.73 |
| | | | | | IRE | | | | |
| ANN | -0.66 | -0.23 | -1.04 | -7.7 | -4.6 | -8.67 | -0.41 | -0.08 | -0.43 |
| DJF | -0.64 | -0.28 | -0.57 | -10.8 | -4.8 | -9.17 | -0.43 | -0.06 | -0.31 |
| MAM | -0.94 | -0.21 | -1.40 | -9.9 | -4.9 | -10.65 | -0.47 | -0.07 | -0.58 |
| JJA | -0.36 | -0.07 | -0.78 | -4.5 | -3.7 | -7.34 | -0.30 | -0.07 | -0.46 |
| SON | -0.70 | -0.38 | -1.38 | -5.7 | -5.1 | -7.52 | -0.45 | -0.13 | -0.37 |

a: Mean over oceanic areas.

b: 27~40°N, 115~123°E.

c: 35~55°N, 140~160°E.