# Peer review of "Seasonal characteristics of emission, distribution, and radiative effect of marine organic aerosols 1 2 over the western Pacific Ocean: an investigation with a coupled regional climate-aerosol model"

_EGUsphere, 2023_

## Author Comment (AC1)

**Dear Editor,**

We are very grateful to the two reviewers for their pertinent comments, careful reading and constructive suggestions, which have helped us to improve the manuscript. We have addressed all their comments carefully and revised the manuscript accordingly by considering their suggestions. Detailed responses to their comments are presented as follows (start with A: in blue font).

**Response to Referee #1:**

This study presents a detailed assessment of marine organic aerosols over the western Pacific Ocean including their radiative effects. This is an interesting and thorough study which highlights the importance of MPOA. I have few comments (especially regarding the radiative calculations) that should be addressed before I can recommend publication.

A: Thank you very much for the valuable comments and constructive suggestions, we carefully address these comments and revised the manuscript accordingly.

**Major comment**

**1. Uncertainty**

Throughout the manuscripts, the authors often provide estimates (model or observations) with 2 to 3 significant digits. This is unwarranted given the uncertainties/errors in both model and observations. The authors need to carefully revise their estimates so that they properly reflect the expected accuracy/precision.

A: Thank you very much for this comment. We check all the observational data and find that the hourly  $PM_{10}$  and  $PM_{2.5}$  observations from EANET and CNEMC have 1 digit after decimal point, the BC and OC observations from the two cruises (Dongfanghong II and KEXUE-1) have 2 digits after decimal point, the OC concentration in previous studies also has 2 digits after decimal point, and AOD from AERONET has at least 3 digits after decimal point.

So, we use the same criterion for all estimates in the revised version, i.e., 2 digits after decimal point for numbers within  $\pm 1.0$  (-1 < numbers < 1) and 1 digit after decimal point for numbers  $\ge 1.0$  or numbers  $\le -1.0$ .

**2. Indirect effect (Section 4.5)**

It's essential that the authors assess whether the estimated DRE and IRE are significant relative to natural variability. In particular, it's likely that the small increases found over land are just natural variability.

A: Thank you very much for this comment. We calculate the significance of perturbations to shortwave radiation induced by MOA and draw the level of significance of DRE and IRE in Figure 10. As shown, the estimated IRE are significant at the 99% confidence level over most areas of the NWP where IRE is the maximum, whereas DRE are significant at the 99% confidence level over only a small area of the NWP (the area of significance for DRE is larger at the the 95% confidence level, figure not shown), we also add descriptions of the level of significance of DRE and IRE in the revised version.

As you point out, the small increases over land could be just natural variability, but we notice that the increase in IRE in some locations over land could also be due to cloud cover change resulting from atmosphere adjustment, that is the increase in IRE is likely related to the decrease in cloud cover.

Line 22. "Apparently" -> be more specific A: Revised.

Fig. 2. It would be interesting to show the contribution of the different components of PM in the model (e.g., sulfate, dust, sea salt, moa, ...). This will help support the discussion of this figure. (See for instance line 371)A: Thank you for the good suggestion. We redraw Fig. 2 by adding mass concentration of major PM10 components, i.e., sulfate, nitrate, ammonium, carbonaceous aerosols, mineral dust, primary PM (directly released PM from

industry, traffic, and construction sites), sea salt, and MOA, along with some discussions in section 3.1 of the revised version.

Fig. 3. I suggest to more clearly distinguish the different sites. At the moment, the figure layout makes it look like a single time series. What is the uncertainty in the observations?

A: Thank you for the good suggestion. We redraw Fig. 3 by dividing it into six small figures for each site. Observations are obtained from EANET (Acid Deposition Monitoring Network in East Asia). The uncertainty in the observation of sodium concentration is less than 5%, according to the EANET report (https://monitoring.eanet.asia/document/public/index).

Fig. 4 The authors highlight improvements in the model performance on 4/12 and 4/10 (Fig. 4b) but it seems MOA degrades the model performance for all other dates between 4/05 and 4/16 (but 4/7). There should be acknowledged A: Thank you for this comment. Yes, the model appears to generally overpredict OC concentrations during 5-16 April over the ocean southeast of Japan, especially on 5-6, 11 and 13 April. The high model biases could be due to potential overpredictions for either land-source OC or marine-source OC, which needs further analysis in the future. We rewrite relevant sentences and clearly point out the model uncertainties in the revised version.

Line 253. What is temporal resolution of biomass burning emissions A: The temporal resolution of biomass burning emissions is daily.

Line 272. Aren't there missing data in the daily Chlorophyll L3 product? How was this handled?

A: Yes, there are missing data in the daily Chlorophyll L3 product mainly due to cloud. To provide as much chlorophyll information as possible for model simulation, for grids without daily data, we used monthly mean data instead. In general, approximately 40~70% daily data are available in most areas of the western Pacific in 2014.

**Line 465. Could SOA from isoprene also be important?**

A: The model simulated SOA concentrations produced by isoprene from land and marine sources are low in this study. Guo et al. (2020) analyzed cruise measurements over the northwestern Pacific Ocean in spring 2014 and found that SOC concentration was less than 8% of total measured OC, and SOC from biogenic VOCs (including both terrestrial and marine sources) only accounted for  $1.5\pm1.4\%$  of total measured OC.

**Reference:**

Guo, T., Guo, Z., Wang, J., Feng, J., Gao, H., and Yao, X.: Tracer-based investigation of organic aerosols in marine atmospheres from marginal seas of China to the northwest Pacific Ocean, Atmos. Chem. Phys., 20, 5055–5070, https://doi.org/10.5194/acp-20-5055-2020, 2020.

Line 482. Please provide the same statistics for open sea vs coastal. It looks like the model is biased low in coastal regions and a little high over the open ocean.

A: Thank you for this suggestion. Below is the daily position of the cruise track in spring 2014. It's difficult to precisely define 'open sea' and 'coastal areas', so we roughly separate 'open sea' (approximately east of 130°E) and 'coastal areas' (cruise track in blue color) in terms of the cruise locations on 3/20 and 4/20. When the cruise was in 'coastal areas' (periods of 3/18-3/20 and 4/20-4/22, 4 observational data were available, see Figure 4b), the observed OC concentration was 3.1  $\mu$ g m-3 on average (1.7~5.1  $\mu$ g m-3) while the simulated average OC concentration was 3.0  $\mu$ g m-3 (1.6~4.0  $\mu$ g m-3). When the cruise was in 'open sea areas' (period of 3/21-4/19, 15 observational data were available, see Figure 4b), the observed and simulated OC concentrations were 0.69  $\mu$ g m-3 (0.04~3.9  $\mu$ g m-3) and 0.65  $\mu$ g m-3 (0.4~1.1  $\mu$ g m-3), respectively.

Figure The locations of the cruise track in March and April 2014

Line 608-610 please provide uncertainty for these estimates

A: The SOA measurements are derived from Guo et al. (2020) and the uncertainties for the measurements are added ( $467\pm384$  ng m-3 over the YBS,  $617\pm649$  ng m-3 over the SCS, and  $155\pm236$  ng m-3 over the NWPO).

Fig. 6 Is it possible to show the contribution of different aerosol types to the overall AOD (at least the MOA)?

A: Yes, we redraw Fig. 6 by adding contributions from MOA, sea salt, and other non-oceanic aerosols (including anthropogenic and other natural aerosols from the Asian continent), which shows the predominant contribution to AOD from non-oceanic aerosols in marine atmosphere, and AOD by sea salt is generally larger than AOD by MOA, consistent with the relative magnitude of DREs by sea salt and MOA. The new figure and relevant discussions are provided in section 3.3 of the revised version.

Line 731-732 The number of significant figures seem unwarranted. It's not clear to me how the comparison between model and measurements is performed.

A: We are sorry for the confusion. We try to compare the simulated marine isoprene emission fluxes from this study with available cruise measurements from previous studies over the western Pacific of East Asia, it is not a real-time comparison. We revise Table S3 for clarity and rewrite relevant descriptions in the revised version.

Line 747. I don't understand this comparison. What are the Arnold emissions for isoprene in the region analyzed here?

A: This comparison is irrelevant; we delete these sentences.

Line 900 "W" missing in front of m-2 A: Sorry for the mistake, it is corrected.

Line 1033: add "-" in front of 0.14 A: Revised.

**Response to Referee #2:**

In "Seasonal characteristics of emission, distribution, and radiative effect of marine organic aerosols over the western Pacific Ocean: an investigation with coupled regional climate-aerosol model," Li et al. examines marine organic aerosol (MOA) over the western Pacific Ocean for the year 2014 using the RIEMS-Chem model. Model is validated against observations, and properties such as MOA emission, concentration, and direct and indirect

radiative effects are examined. The study is comprehensive and thorough, and publication can be advised after addressing the following comments.

Main comments:

- Please revise the descriptions for determining the radiative effects (comments 16, 17, and 23 below) and the interpretation of atmospheric adjustments (comments 25 and 26 below).

- Have cloud properties been validated for this model before? How pristine are the remote marine clouds, particularly where strong IRE\_MOA are found? (comment 18 below)

- Please provide more discussion on why IRE\_MOA is greater than IRE\_sea salt, and why IRE\_sea salt is weaker than DRE\_sea salt (comment 21 below).

A: Thank you very much for the valuable comments and constructive suggestions, we carefully address these comments and revised the manuscript accordingly.

**All comments:**

1. In the abstract, lines 26-29, please clarify if the max and min seasons refer to MOA emission or concentration. A: Revised. The max and min seasons refer to MPOA concentration.

2. Lines 94-98: Should this information be in the methods section instead of the introduction?

A: Thank you for this suggestion. We move this information to section 2.3 in the the revised version.

3. Please define the boundaries (e.g., latitude/longitude bounds) of the model domain as well as for EYB and NWP in the main text.

A: Thank you for this suggestion. We add latitude/longitude bounds of the model domain and the subregions of EYB and NWP when they first appear in the main text in the revised version.

4. What is the model resolution and time step?

A: The model resolution is 60 km and the time step is 180 seconds.

5. Line 711: negative correlation between \_\_\_\_(?) and sea salt emission

A: Sorry for the mistake. It is revised to "which could be due to the slight negative correlation between Chl-a and sea salt emission in MAM".

6. Line 735: Where are equations (2) and (3) defined?A: They are defined in the supplement. We add this information in the revised version.

7. Line 767: How far inland does "inland areas" refer to?

A: Approximately 600 km away from the coastline of east China according to Figure 8a.

8. Is 2014 a typical and representative year in terms of meteorology in the western Pacific region? Please discuss this particularly in the context of modelled MOA emission and concentration results in pages 27-28.

A: It appears that 2014 is a normal year (neither El Niño nor La Niña) which can be reflected by the ENSO index (https://psl.noaa.gov/enso/mei/). The main reason we choose 2014 as the study period is the availability of cruise campaigns (for model validation and analysis) over the East China Sea and the western Pacific from spring to early summer in 2014, when BC and OC concentrations were measured. We add relevant descriptions in the revised version.

9. Line 814: Perhaps rewrite the sentence starting with "Furthermore" as "Furthermore, SON has the second highest average precipitation in the NWP (Table 9), which caused... and contributed to..."A: Thank you very much. We rewrite this sentence in the revision.

10. Line 820: Please specify "within the model domain" after "latitudes" A: Revised.

11. Line 853: southern parts of "the model domain" in the western Pacific A: Revised

12. Please clarify whether land grids are included when regional (e.g. for NWP and FYB) averages are quoted A: Land grids are not included in the regional averages for NWP and EYB. We clarify this in the revised version.

13. Line 894: Please clarify how the 10% difference in relative humidity may affect the DREA: Higher relative humidity favors hygroscopic growth of MOA, leading to larger light extinction coefficients, and thus stronger DRE. We describe the processes in the revised version.

14. Line 903-907: Any insights on why the seasonal trend in EYB is different from that in NWP? A: The different seasonal trends of DRE in EYB and NWP could be mainly attributed to the different seasonality of MOA concentrations in the two regions. We add the explanation for the different trend in the revised version.

15. Line 912: Have sea salt and anthropogenic aerosol been validated in this model before? If not, please add "in this simulation" after "Pacific" in line 913

A: Sea salt and anthropogenic aerosol concentrations are partly validated at EANET and CNEMC sites in this study (Figure 2 for  $PM_{10}$  concentrations, Figure 3 for sodium concentration) with discussions in section 3.1. We add "in this simulation" after "Pacific" in the revision.

16. Instead of two radiation calls to calculate IRE as stated on line 936, is it not the case that the whole calculation for cloud properties as outlined in lines 946 to 950 is called twice? If yes, please amend as necessary.

A: Yes, cloud properties (e.g., cloud optical depth, cloud single scattering albedo, cloud asymmetry factor) is called twice in IRE calculation, one with cloud properties due to total aerosols (explained in the response to question 22) and one with those due to total aerosols but MOA, the difference between the two calls reflects the instantaneous perturbation of MOA to cloud and radiation. We explain more clearly the IRE and DRE calculations in the revised version.

17. Line 941: Does this equation not also apply for DRE? It doesn't seem particularly helpful in differentiating DRE and IRE calculations.

A: Yes, this equation also applies for DRE, so we move it to the description of DRE calculation in the revised version.

18. How pristine are the marine clouds in the model (e.g. median Nc and drop radius)? Both before and after addition of MOA. How does it compare to observations -- does it provide a realistic background condition on which MOA's impact can be determined?

A: Thank you for this comment. In this study, the background/pristine marine cloud condition is represented by prescribing the lowest bound of Nc of 10/cm3, which generally represent liquid stratiform cloud in clean marine condition according to previous cruise and satellite observations (Yum et al., 1998; Bower et al., 2006; Hoose et al.,

2009; Zeng et al., 2014). We are aware that the prescription of low bound of Nc may affect the estimation of IRE as pointed out by Hoose et al. (2009), a higher low bound of Nc (e.g., 20/cm3 or more) will lead to lower estimation of IRE, we clearly pointed out this uncertainty in the revised version.

Cloud properties modeling is one of the most challenging works in climate simulation/prediction. As your suggestion, we try to collect MODIS retrievals for cloud amount and cloud optical depth and compare with simulated cloud properties in the presence of MOA, sea salt and anthropogenic aerosols as shown below. In general, the model is able to capture the major distribution features of cloud properties, despite the model tends to underpredict cloud optical depth over the northern parts of the study domain. Such model biases are often found in current meteorological or climate modeling, e.g., CESM/CAM5 (Gantt et al., 2014), WRF-Chem (Wang et al., 2015), and the uncertainties in satellite retrievals could also lead to the observation-model biases. In general, the simulated cloud properties in this study are within the acceptable range and of similar skill to other climate models. We add a brief description on the comparison between model simulations and MODIS retrievals in the revised version.

---

## Author Comment (AC2)

Dear Editor,

We are very grateful to the two reviewers for their pertinent comments, careful reading and constructive suggestions, which have helped us to improve the manuscript. We have addressed all their comments carefully and revised the manuscript accordingly by considering their suggestions. Detailed responses to their comments are presented as follows (start with A: in blue font).

***Response to Referee #1:***

This study presents a detailed assessment of marine organic aerosols over the western Pacific Ocean including their radiative effects. This is an interesting and thorough study which highlights the importance of MPOA. I have few comments (especially regarding the radiative calculations) that should be addressed before I can recommend publication.

A: Thank you very much for the valuable comments and constructive suggestions, we carefully address these comments and revised the manuscript accordingly.

Major comment

1. Uncertainty

Throughout the manuscripts, the authors often provide estimates (model or observations) with 2 to 3 significant digits. This is unwarranted given the uncertainties/errors in both model and observations. The authors need to carefully revise their estimates so that they properly reflect the expected accuracy/precision.

A: Thank you very much for this comment. We check all the observational data and find that the hourly $PM_{10}$ and $PM_{2.5}$ observations from EANET and CNEMC have 1 digit after decimal point, the BC and OC observations from the two cruises (Dongfanghong II and KEXUE-1) have 2 digits after decimal point, the OC concentration in previous studies also has 2 digits after decimal point, and AOD from AERONET has at least 3 digits after decimal point.

So, we use the same criterion for all estimates in the revised version, i.e., 2 digits after decimal point for numbers within ±1.0 (-1 < numbers < 1) and 1 digit after decimal point for numbers ≥ 1.0 or numbers ≤ -1.0.

2. Indirect effect (Section 4.5)

It's essential that the authors assess whether the estimated DRE and IRE are significant relative to natural variability. In particular, it's likely that the small increases found over land are just natural variability.

A: Thank you very much for this comment. We calculate the significance of perturbations to shortwave radiation induced by MOA and draw the level of significance of DRE and IRE in Figure 10. As shown, the estimated IRE are significant at the 99% confidence level over most areas of the NWP where IRE is the maximum, whereas DRE are significant at the 99% confidence level over only a small area of the NWP (the area of significance for DRE is larger at the the 95% confidence level, figure not shown), we also add descriptions of the level of significance of DRE and IRE in the revised version.

As you point out, the small increases over land could be just natural variability, but we notice that the increase in IRE in some locations over land could also be due to cloud cover change resulting from atmosphere adjustment, that is the increase in IRE is likely related to the decrease in cloud cover.

Line 22. "Apparently" -> be more specific

A: Revised.

Fig. 2. It would be interesting to show the contribution of the different components of PM in the model (e.g., sulfate, dust, sea salt, moa, …). This will help support the discussion of this figure. (See for instance line 371)

A: Thank you for the good suggestion. We redraw Fig. 2 by adding mass concentration of major $PM_{10}$ components, i.e., sulfate, nitrate, ammonium, carbonaceous aerosols, mineral dust, primary PM (directly released PM from

industry, traffic, and construction sites), sea salt, and MOA, along with some discussions in section 3.1 of the revised version.

Fig. 3. I suggest to more clearly distinguish the different sites. At the moment, the figure layout makes it look like a single time series. What is the uncertainty in the observations?
A: Thank you for the good suggestion. We redraw Fig. 3 by dividing it into six small figures for each site. Observations are obtained from EANET (Acid Deposition Monitoring Network in East Asia). The uncertainty in the observation of sodium concentration is less than 5%, according to the EANET report (https://monitoring.eanet.asia/document/public/index).

Fig. 4 The authors highlight improvements in the model performance on 4/12 and 4/10 (Fig. 4b) but it seems MOA degrades the model performance for all other dates between 4/05 and 4/16 (but 4/7). There should be acknowledged
A: Thank you for this comment. Yes, the model appears to generally overpredict OC concentrations during 5-16 April over the ocean southeast of Japan, especially on 5-6, 11 and 13 April. The high model biases could be due to potential overpredictions for either land-source OC or marine-source OC, which needs further analysis in the future. We rewrite relevant sentences and clearly point out the model uncertainties in the revised version.

Line 253. What is temporal resolution of biomass burning emissions
A: The temporal resolution of biomass burning emissions is daily.

Line 272. Aren't there missing data in the daily Chlorophyll L3 product? How was this handled?
A: Yes, there are missing data in the daily Chlorophyll L3 product mainly due to cloud. To provide as much chlorophyll information as possible for model simulation, for grids without daily data, we used monthly mean data instead. In general, approximately 40~70% daily data are available in most areas of the western Pacific in 2014.

Line 465. Could SOA from isoprene also be important?
A: The model simulated SOA concentrations produced by isoprene from land and marine sources are low in this study. Guo et al. (2020) analyzed cruise measurements over the northwestern Pacific Ocean in spring 2014 and found that SOC concentration was less than 8% of total measured OC, and SOC from biogenic VOCs (including both terrestrial and marine sources) only accounted for 1.5±1.4% of total measured OC.

Reference:
Guo, T., Guo, Z., Wang, J., Feng, J., Gao, H., and Yao, X.: Tracer-based investigation of organic aerosols in marine atmospheres from marginal seas of China to the northwest Pacific Ocean, Atmos. Chem. Phys., 20, 5055–5070, https://doi.org/10.5194/acp-20-5055-2020, 2020.

Line 482. Please provide the same statistics for open sea vs coastal. It looks like the model is biased low in coastal regions and a little high over the open ocean.
A: Thank you for this suggestion. Below is the daily position of the cruise track in spring 2014. It's difficult to precisely define 'open sea' and 'coastal areas', so we roughly separate 'open sea' (approximately east of 130°E) and 'coastal areas' (cruise track in blue color) in terms of the cruise locations on 3/20 and 4/20. When the cruise was in 'coastal areas' (periods of 3/18-3/20 and 4/20-4/22, 4 observational data were available, see Figure 4b), the observed OC concentration was 3.1 μg m$^{-3}$ on average (1.7~5.1 μg m$^{-3}$) while the simulated average OC concentration was 3.0 μg m$^{-3}$ (1.6~4.0 μg m$^{-3}$). When the cruise was in 'open sea areas' (period of 3/21-4/19, 15 observational data were available, see Figure 4b), the observed and simulated OC concentrations were 0.69 μg m$^{-3}$ (0.04~3.9 μg m$^{-3}$) and 0.65 μg m$^{-3}$ (0.4~1.1 μg m$^{-3}$), respectively.

[Figure]

Figure The locations of the cruise track in March and April 2014

Line 608-610 please provide uncertainty for these estimates
A: The SOA measurements are derived from Guo et al. (2020) and the uncertainties for the measurements are added ($467\pm384$ ng m$^{-3}$ over the YBS, $617\pm649$ ng m$^{-3}$ over the SCS, and $155\pm236$ ng m$^{-3}$ over the NWPO).

Fig. 6 Is it possible to show the contribution of different aerosol types to the overall AOD (at least the MOA)?
A: Yes, we redraw Fig. 6 by adding contributions from MOA, sea salt, and other non-oceanic aerosols (including anthropogenic and other natural aerosols from the Asian continent), which shows the predominant contribution to AOD from non-oceanic aerosols in marine atmosphere, and AOD by sea salt is generally larger than AOD by MOA, consistent with the relative magnitude of DREs by sea salt and MOA. The new figure and relevant discussions are provided in section 3.3 of the revised version.

Line 731-732 The number of significant figures seem unwarranted. It's not clear to me how the comparison between model and measurements is performed.
A: We are sorry for the confusion. We try to compare the simulated marine isoprene emission fluxes from this study with available cruise measurements from previous studies over the western Pacific of East Asia, it is not a real-time comparison. We revise Table S3 for clarity and rewrite relevant descriptions in the revised version.

Line 747. I don't understand this comparison. What are the Arnold emissions for isoprene in the region analyzed here?
A: This comparison is irrelevant; we delete these sentences.

Line 900 "W" missing in front of m-2
A: Sorry for the mistake, it is corrected.

Line 1033: add "-" in front of 0.14
A: Revised.

***Response to Referee #2:***
In "Seasonal characteristics of emission, distribution, and radiative effect of marine organic aerosols over the western Pacific Ocean: an investigation with coupled regional climate-aerosol model," Li et al. examines marine organic aerosol (MOA) over the western Pacific Ocean for the year 2014 using the RIEMS-Chem model. Model is validated against observations, and properties such as MOA emission, concentration, and direct and indirect

radiative effects are examined. The study is comprehensive and thorough, and publication can be advised after addressing the following comments.

Main comments:
- Please revise the descriptions for determining the radiative effects (comments 16, 17, and 23 below) and the interpretation of atmospheric adjustments (comments 25 and 26 below).
- Have cloud properties been validated for this model before? How pristine are the remote marine clouds, particularly where strong IRE_MOA are found? (comment 18 below)
- Please provide more discussion on why IRE_MOA is greater than IRE_sea salt, and why IRE_sea salt is weaker than DRE_sea salt (comment 21 below).
A: Thank you very much for the valuable comments and constructive suggestions, we carefully address these comments and revised the manuscript accordingly.

All comments:
1. In the abstract, lines 26-29, please clarify if the max and min seasons refer to MOA emission or concentration.
A: Revised. The max and min seasons refer to MPOA concentration.

2. Lines 94-98: Should this information be in the methods section instead of the introduction?
A: Thank you for this suggestion. We move this information to section 2.3 in the the revised version.

3. Please define the boundaries (e.g., latitude/longitude bounds) of the model domain as well as for EYB and NWP in the main text.
A: Thank you for this suggestion. We add latitude/longitude bounds of the model domain and the subregions of EYB and NWP when they first appear in the main text in the revised version.

4. What is the model resolution and time step?
A: The model resolution is 60 km and the time step is 180 seconds.

5. Line 711: negative correlation between _____(?) and sea salt emission
A: Sorry for the mistake. It is revised to "which could be due to the slight negative correlation between Chl-a and sea salt emission in MAM".

6. Line 735: Where are equations (2) and (3) defined?
A: They are defined in the supplement. We add this information in the revised version.

7. Line 767: How far inland does "inland areas" refer to?
A: Approximately 600 km away from the coastline of east China according to Figure 8a.

8. Is 2014 a typical and representative year in terms of meteorology in the western Pacific region? Please discuss this particularly in the context of modelled MOA emission and concentration results in pages 27-28.
A: It appears that 2014 is a normal year (neither El Niño nor La Niña) which can be reflected by the ENSO index (https://psl.noaa.gov/enso/mei/). The main reason we choose 2014 as the study period is the availability of cruise campaigns (for model validation and analysis) over the East China Sea and the western Pacific from spring to early summer in 2014, when BC and OC concentrations were measured. We add relevant descriptions in the revised version.

9. Line 814: Perhaps rewrite the sentence starting with "Furthermore" as "Furthermore, SON has the second highest average precipitation in the NWP (Table 9), which caused... and contributed to..."
A: Thank you very much. We rewrite this sentence in the revision.

10. Line 820: Please specify "within the model domain" after "latitudes"
A: Revised.

11. Line 853: southern parts of "the model domain" in the western Pacific
A: Revised

12. Please clarify whether land grids are included when regional (e.g. for NWP and FYB) averages are quoted
A: Land grids are not included in the regional averages for NWP and EYB. We clarify this in the revised version.

13. Line 894: Please clarify how the 10% difference in relative humidity may affect the DRE
A: Higher relative humidity favors hygroscopic growth of MOA, leading to larger light extinction coefficients, and thus stronger DRE. We describe the processes in the revised version.

14. Line 903-907: Any insights on why the seasonal trend in EYB is different from that in NWP?
A: The different seasonal trends of DRE in EYB and NWP could be mainly attributed to the different seasonality of MOA concentrations in the two regions. We add the explanation for the different trend in the revised version.

15. Line 912: Have sea salt and anthropogenic aerosol been validated in this model before? If not, please add "in this simulation" after "Pacific" in line 913
A: Sea salt and anthropogenic aerosol concentrations are partly validated at EANET and CNEMC sites in this study (Figure 2 for $PM_{10}$ concentrations, Figure 3 for sodium concentration) with discussions in section 3.1. We add "in this simulation" after "Pacific" in the revision.

16. Instead of two radiation calls to calculate IRE as stated on line 936, is it not the case that the whole calculation for cloud properties as outlined in lines 946 to 950 is called twice? If yes, please amend as necessary.
A: Yes, cloud properties (e.g., cloud optical depth, cloud single scattering albedo, cloud asymmetry factor) is called twice in IRE calculation, one with cloud properties due to total aerosols (explained in the response to question 22) and one with those due to total aerosols but MOA, the difference between the two calls reflects the instantaneous perturbation of MOA to cloud and radiation. We explain more clearly the IRE and DRE calculations in the revised version.

17. Line 941: Does this equation not also apply for DRE? It doesn't seem particularly helpful in differentiating DRE and IRE calculations.
A: Yes, this equation also applies for DRE, so we move it to the description of DRE calculation in the revised version.

18. How pristine are the marine clouds in the model (e.g. median Nc and drop radius)? Both before and after addition of MOA. How does it compare to observations -- does it provide a realistic background condition on which MOA's impact can be determined?
A: Thank you for this comment. In this study, the background/pristine marine cloud condition is represented by prescribing the lowest bound of Nc of $10/cm^3$, which generally represent liquid stratiform cloud in clean marine condition according to previous cruise and satellite observations (Yum et al., 1998; Bower et al., 2006; Hoose et al.,

2009; Zeng et al., 2014). We are aware that the prescription of low bound of Nc may affect the estimation of IRE as pointed out by Hoose et al. (2009), a higher low bound of Nc (e.g., 20/cm$^3$ or more) will lead to lower estimation of IRE, we clearly pointed out this uncertainty in the revised version.

Cloud properties modeling is one of the most challenging works in climate simulation/prediction. As your suggestion, we try to collect MODIS retrievals for cloud amount and cloud optical depth and compare with simulated cloud properties in the presence of MOA, sea salt and anthropogenic aerosols as shown below. In general, the model is able to capture the major distribution features of cloud properties, despite the model tends to underpredict cloud optical depth over the northern parts of the study domain. Such model biases are often found in current meteorological or climate modeling, e.g., CESM/CAM5 (Gantt et al., 2014), WRF-Chem (Wang et al., 2015), and the uncertainties in satellite retrievals could also lead to the observation-model biases. In general, the simulated cloud properties in this study are within the acceptable range and of similar skill to other climate models. We add a brief description on the comparison between model simulations and MODIS retrievals in the revised version.

[Figure]

Figure MODIS retrieved (a,c) and model simulated (b,d) annual mean cloud fraction (a,b) and cloud optical depth (c,d)

References

Bower, K., T. Choularton, J. Latham, J. Sahraei, and S. Salter, 2006. Computational assessment of a proposed technique for global warming mitigation via albedo-enhancement of marine stratocumulus clouds, Atmos. Res., 82, 328 – 336.

Gantt, B., He, J., Zhang, X., Zhang, Y., Nenes, A., 2014. Incorporation of advanced aerosol activation treatments into CESM/CAM5: model evaluation and impacts on aerosol indirect effects. Atmos. Chem. Phys. 14, 7485–7497.

Zeng, S., Riedi, J., Trepte, C. R., Winker, D. M., and Hu, Y. X., 2014. Study of global cloud droplet number concentration with A-Train satellites. Atmos. Chem. Phys. 14, 7125–7134.

Wang, K., Zhang, Y., Yahya, K., Wu, S.-Y., Grell, G., 2015. Implementation and initial application of new chemistry-aerosol options in WRF/Chem for simulating secondary organic aerosols and aerosol indirect effects for regional air quality. Atmos. Environ. 115, 716–732.

Yum, S. S., Hudson, J. G., and Xie, Y., 1998. Comparisons of cloud microphysics with cloud condensation nuclei spectra over the summertime Southern Ocean, J. Geophys. Res., 103, 16,625 – 16,636.

19. Line 963 and figure 10 in general: How is the relationship between the spatial distribution of IRE and that of clouds?

A: The distribution of IRE depends on both MOA concentration and cloud distribution, so the spatial distribution of IRE is not fully consistent with that of cloud. As shown in Figure 10h, the strongest IRE occurs over the NWP (northeastern part of the domain) mainly due to higher Chl-a and MOA concentration and moderate cloud amount there, although the maximum cloud amount occurs over the southern part of the domain (Figure S3h).

20. Line 964: Which regions?

A: The regions are from the East China Sea to the oceans east of Japan. We revise the sentence in the revision.

21. Line 1011: Any speculations on why IRE_MOA is generally stronger than IRE_sea salt in this study? Sea salt mass and **number** should both be greater than MOA (is it?). Is it a matter of size? Hygroscopicity? Also, why is IRE_sea salt weaker than DRE_sea salt, in contrast to both MOA and anthropogenic aerosol?

A: The number concentration of MPOA and sea salt is calculated with the following equation:

$$N_i = \frac{M_i}{\frac{4}{3}\pi r_i^3 \rho_i \exp\left(\frac{9}{2}\log^2\sigma_i\right)}$$

where $N_i$ and $M_i$ are the number and mass concentrations, $r_i$ and $\sigma_i$ are the geometric mean dry radius and standard deviation of aerosol component i, all aerosol chemical components are assumed to be log-normal distribution.

The typical radius of MOA is set to 0.05 μm according to cruise measurements over the western Pacific (Feng et al., 2017), while the size distribution of sea salt is divided into two modes with the mean fine mode radius being 0.1μm and the mean coarse mode radius being 1 μm according to measurements from Gong et al. (1997). Because coarse sea salt is easily deposited to sea surface, fine sea salt dominates total number of sea salt. Although the hygroscopicity of sea salt is larger than that of MOA, because the radius of MOA is smaller than the radius of fine sea salt, the number concentration of MOA is higher than that of sea salt according to the above equation (given the same σI of 1.6), thus the activated cloud droplet number concentration by MOA is larger than that by fine sea salt as shown in the following Figure, which can explain why the $IRE_{MOA}$ is larger than $IRE_{sea}$ in this study.

[Figure]

Figure Model simulated annual mean activated cloud droplet number concentration by MOA (a) and by sea salt (b)

The reason why IRE_sea salt is weaker than DRE_sea salt in contrast to both MOA and anthropogenic aerosol could be explained by the relative magnitude of DRE efficiency of these aerosols, which is defined as DRE/AOD (unit: W m$^{-2}$ τ$^{-1}$). The following table presents the model calculated DRE efficiency of each type of aerosols, which clearly shows that the efficiency of sea salt is remarkably larger than those of MOA and anthropogenic aerosols over the western Pacific, which indicates that sea salt tends to exert a stronger DRE than other aerosols and could exceed IRE by sea salt. It is noted that there is no consistent results on the relative magnitude of DRE and IRE of

sea salt in previous global modeling studies. Paulot et al. (2020) estimated a stronger global mean sea salt IRE (-1.2 W m$^{-2}$) than DRE (-0.8 W m$^{-2}$), but in some other studies, model results are opposite (Ayash et al., 2008; Rap et al., 2013). Ayash et al. (2008) estimated a stronger DRE of sea salt (-0.65 W m$^{-2}$) than IRE (-0.38 W m$^{-2}$) in terms of global and annual mean.

Model simulated DRE efficiency of various types of aerosols over different areas of the western Pacific

|  | MOA | | | Anthropogenic | | | Sea salt | | |
|---|---|---|---|---|---|---|---|---|---|
|  | WP[a] | EYB[b] | NWP[c] | WP[a] | EYB[b] | NWP[c] | WP[a] | EYB[b] | NWP[c] |
|  | DRE/AOD | | | | | | | | |
| ANN | -17 | -15 | -18 | -19 | -25 | -15 | -31 | -33 | -26 |

References

Ayash, T., Gong, S., Jia, C.Q.: Direct and indirect shortwave radiative effects of sea salt aerosols. J. Clim, 21, 3207–3220, doi:10.1175/2007JCLI2063.1, 2008.

Feng, L.M., Shen, H.Q., Zhu, Y.J., Gao, H.W., and Yao, X.H.: Insight into Generation and Evolution of Sea-Salt Aerosols from Field Measurements in Diversified Marine and Coastal Atmospheres, Sci. Rep., 7, 41260; doi: 10.1038/srep41260, 2017.

Gong, S. L., L. A. Barrie, and J.-P. Blanchet: Modeling sea-salt aerosols in the atmosphere 1. Model development, J. Geophys. Res., 102(D3), 3805-3818, doi:10.1029/96JD02953, 1997.

Rap, A., Scott, C.E., Spracklen, D.V., Bellouin, N., Forster, P.M., Carslaw, K.S., Schmidt, A., and Mann, G.: Natural aerosol direct and indirect radiative effects, Geophys. Res. Lett., 40, 3297-3301, doi:10.1002/grl.50441, 2013.

Paulot, F., Paynter, D., Winton, M., Ginoux, P., Zhao, M., Horowitz, L. W.: Revisiting the impact of sea salt on climate sensitivity, Geophys. Res. Lett., 47, e2019GL085601, doi:10.1029/2019GL085601, 2020

22. Line 1041 and other instances: Please don't state "all aerosols" if only MOA, sea salt, and anthropogenic aerosol are included, to the exclusion of dust, biogenic OCs, etc.
A: Thank you for this suggestion. This model includes primary and secondary anthropogenic aerosols, mineral dust, SOA produced by biogenic VOCs of land and ocean, primary and secondary aerosols from wildfire which are described in section 2.1 and 2.3, but it does not include some natural aerosols such as primary OC from vegetation, fugitive dust, etc., we admit that it is not appropriate to state "all aerosols", so we use "total aerosols" instead of "all aerosols" for brevity, and we clearly describe that total aerosols include " sea spray (sea salt+MOA), anthropogenic aerosols and other natural aerosols (mineral dust, SOA from vegetation emitted VOCs) considered in this model" in the revised version.

23. Please describe early on and more clearly in section 4.5 what simulations are used to determine DRE_ari and IRE_aci. Firstly, to make clear that these are not the same simulations as those used to calculate DRE_MOA and IRE_MOA, and secondly, to clarify that the ones used to determine DRE_ari are not the same as the ones for IRE_aci (which is currently implied in line 1066: "between the two simulations").
A: Thank you very much for this suggestion. We describe more clearly the calculation method for DREari and IREaci in section 4.5 in the revised version as "Note the method for calculating $DRE_{ari}$ and $IRE_{aci}$ is different from that for $DRE_{MOA}$ and $IRE_{MOA}$. $DRE_{ari}$ is derived by the difference in shortwave radiation flux at TOA induced by MOA scattering between two simulations with aerosol optical properties due to total aerosols and with those due to total aerosols but MOA, while the perturbation of MOA to cloud properties is turned off. $IRE_{aci}$ is calculated by the difference in shortwave radiation flux at TOA induced by MOA perturbation to cloud albedo between two

simulations with cloud properties due to total aerosols and with those due to total aerosols but MOA, while the direct perturbation of MOA to radiation is closed. The calculation of DRE$_{ari}$ and IRE$_{aci}$ considers the adjustment of atmospheric temperature, water vapor, and cloud (including cloud microphysical and lifetime change, i.e., the second indirect effect) to the MOA-induced radiation perturbation in the two simulations."

24. Lines 1073 and 1075: W m-2 instead of W m-3
A: Revised.

25. Line 1074: Comments relating to the atmospheric response and adjustment: Should this not be examined from comparison between (DRE_ari-DRE_MOA) and (IRE_aci-IRE_MOA)?
A: Yes, it could be clearer to show the atmospheric response and adjustment induced by DRE$_{MOA}$ and IRE$_{MOA}$ through drawing (DRE_ari-DRE_MOA) and (IRE_aci-IRE_MOA), but considering the difference is not so large and to reduce the number of figures, we describe the difference just by comparing Figure 10a and Figure 11a, and Figure 10f and Figure 11b.

26. Relating to the above, for line 1074 and line 1079, it's not just the positive values. Fast response and adjustments may alter the magnitude of the radiative effects without resulting in positive values.
A: Thank you for this comment, yes, both magnitude and sign of radiative effects could be altered due to fast response and adjustments, we delete the inappropriate sentence in the revision.

27. Line 1128: "all" --> "sea spray and anthropogenic"
A: Revised to "total aerosols" as explained above.

28. Table 1 and in general for model validation: Are the simulation values that of the model grid containing the observation site? Please specify in the methods description or beginning of results.
A: Model results are extracted from the model grid closest to the observational site for comparison. We describe the method for model-observations comparison in the revised version.

---

## Author Response (AR2)

Dear Editor,

  Thank you very much for handing this manuscript. We are very grateful to you and the two reviewers for the positive comments on our response and revision. We have addressed the additional comments raised by the referee #2 and revised the manuscript accordingly as follows (start with A: in blue font).

***Response to Referee #2***:

- Regarding comment 4 in the previous round of revisions, please add the model resolution and time step to section 2.4 on model setup in the manuscript.

A: Added in line 291 of the revision.

- It may also be worthwhile to mention in the manuscript some of your answer to comment 21 previously, on the difference in size and number concentrations of MOA and sea salt (just a sentence or two perhaps), and/or include the discussion in the supplement.

A: Thank you very much for the good suggestion. We add several sentences (in line 1034-1037 of section 4.4) to explain why $IRE_{MOA}$ is stronger than $IRE_{sea\ salt}$ as "Although the hygroscopicity of sea salt is larger than that of MOA, because the mean radius of MOA (0.05 $\mu$m) is smaller than the radius of sea salt (0.1 $\mu$m for fine mode), the number concentration of MOA is larger than that of sea salt, leading to larger activated cloud droplet number concentration and IRE by MOA than those by sea salt" in the revised version.

- Thanks for clarifying the aerosol species included. It is fine to call it "all aerosols" if all aerosol species in the model are included (the previous version just implied that mineral dust and other natural aerosol in the model were not included in the sum), though "total aerosols" is also fine.

A: Thank you for the comment, we would like to use "total aerosols" in the revised version.

- An additional comment regarding figure 11: for clarity, please state "DRE_ari" and "IRE_aci" in the figure caption as well.

A: Added.